# ROBUSTNESS INSPIRED GRAPH BACKDOOR DEFENSE

**Zhiwei Zhang**   **Minhua Lin**   **Junjie Xu**   **Zongyu Wu**   **Enyan Dai**   **Suhang Wang**
The Pennsylvania State University
`{zbz5349, mfl5681, junjiexu, zongyuwu, emd5759, szw494}@psu.edu`

## ABSTRACT

Graph Neural Networks (GNNs) have achieved promising results in tasks such as node classification and graph classification. However, recent studies reveal that GNNs are vulnerable to backdoor attacks, posing a significant threat to their real-world adoption. Despite initial efforts to defend against specific graph backdoor attacks, there is no work on defending against various types of backdoor attacks where generated triggers have different properties. Hence, we first empirically verify that prediction variance under edge dropping is a crucial indicator for identifying poisoned nodes. With this observation, we propose using random edge dropping to detect backdoors and theoretically show that it can efficiently distinguish poisoned nodes from clean ones. Furthermore, we introduce a novel robust training strategy to efficiently counteract the impact of the triggers. Extensive experiments on real-world datasets show that our framework can effectively identify poisoned nodes, significantly degrade the attack success rate, and maintain clean accuracy when defending against various types of graph backdoor attacks with different properties. Our code is available at: github.com/zzwjames/RIGBD.

## 1 INTRODUCTION

Graphs are pervasive in real world such as social networks (Fan et al., 2019), molecular graphs (Mansimov et al., 2019), and knowledge graphs (Liu et al., 2022). Graph Neural Networks (GNNs) have shown great ability in node representation learning on graphs. Generally, GNNs adopt a message-passing mechanism that updates a node's representation by iteratively aggregating information from its neighbors. The resulting node representations retain both node attributes and local graph structure information, benefiting various downstream tasks such as node classification (Hamilton et al., 2017; Kipf & Welling, 2016; Velickovic et al., 2017) and graph classification (Xu et al., 2018).

Despite their great performance, recent studies (Dai et al., 2023; Xi et al., 2021; Zhang et al., 2021; 2024) show that GNNs are susceptible to backdoor attacks. Generally, backdoor attacks involve generating and attaching backdoor triggers to a small set of target nodes, which are then assigned a target class. These triggers can be predefined or created by a trigger generator and usually take the form of a node or subgraph. When a GNN model is trained on a backdoored dataset, it learns to associate the presence of the trigger with the target class. Consequently, during inference, the backdoored model will misclassify test nodes with the trigger attached as belonging to the target class, while still maintaining high accuracy on clean nodes without a trigger attached. Backdoor attacks on GNNs are particularly concerning due to their potential impact on critical real-world applications such as healthcare and financial services. For instance, in medical diagnosis networks, an attacker could embed backdoor triggers into the training data, leading the GNN model to misclassify certain medical conditions when the trigger is present.

Thus, graph backdoor attack and defense are attracting increasing attention and several initial efforts have been taken (Xi et al., 2021; Zhang et al., 2021; 2024; Dai et al., 2023). For example, the seminal work SBA (Zhang et al., 2021) adopts randomly generated graphs as triggers for graph backdoor attack. To improve the attack success rate, GTA (Xi et al., 2021) adopts a backdoor trigger generator to generate more powerful sample-specific triggers. Observing that the backdoor trigger and target nodes tend to have low feature similarity, Dai et al. (2023) propose a defense method called Prune, which removes edges that connect nodes with low cosine similarity. It significantly reduces the attack success rate (ASR) of previous works. To improve the stealthiness of backdoor attacks, they proposed unnoticeable backdoor attack method UGBA by maximizing the cosine similarity between backdoor

triggers and target nodes. Zhang et al. (2024) show that backdoor triggers tend to be outliers which can be removed with graph outlier detection (OD). To address this, they further proposed DPGBA which can generate in-distribution triggers. Despite initial efforts on backdoor defense, they generally utilize backdoor specific properties to defend against specific backdoor and are ineffective across various types of backdoor triggers and attack methods.

Therefore, in this paper, we study an important problem of developing an effective graph backdoor defense method against various types of backdoor triggers and attack methods. In essence, we are faced with two challenges: **(i)** How to efficiently and precisely identify poisoned nodes and backdoor triggers, even when those triggers are indistinguishable from clean nodes? **(ii)** How to minimize the impact of backdoor triggers when some of the triggers are not identified? In an attempt to address these challenges, we propose a novel framework Robustness Inspired Graph Backdoor Defense (RIGBD). To efficiently and precisely identify poisoned nodes, we empirically show in Section 3.2 that removing edges linking backdoor triggers typically leads to large prediction variance for poisoned target nodes. Based on this observation, we propose training a backdoored model with specially designed graph convolution operations on a poisoned graph, performing random edge dropping, and identifying nodes with high prediction variance as poisoned nodes. With candidate poisoned nodes and identified target class, we propose a novel robust GNN training loss, which minimizes model's prediction confidence on the target class for poisoned nodes to efficiently counteract the impact of the triggers. Such strategy is effective even if part of poisoned nodes are not identified in the training set.

Our **main contributions** are: **(i)** We empirically verify that poisoned nodes typically exhibit large prediction variance under edge dropping. **(ii)** Theoretical analysis guarantees that our specially designed graph convolution operations can precisely distinguish poisoned nodes from clean nodes through random edge dropping. **(iii)** We propose a novel training strategy to train a backdoor robust GNN model even though some poisoned nodes are not identified. **(iv)** Extensive experiments show the effectiveness of RIGBD in defending against backdoor attacks and maintaining clean accuracy.

## 2 RELATED WORK

**Graph Backdoor Attacks.** SBA (Zhang et al., 2021) is a seminal work on graph backdoor attacks, which adopts randomly generated graphs as triggers. GTA (Xi et al., 2021) adopts a backdoor trigger generator to generate more powerful sample-specific triggers to improve the attack success rate. UGBA (Dai et al., 2023) introduces an unnoticeable loss function aimed at maximizing the cosine similarity between backdoor triggers and target nodes to improve the stealthiness of their attack. DPGBA (Zhang et al., 2024) introduces an outlier detector and uses adversarial learning to generate in-distribution triggers, addressing low ASR or outlier issues in existing graph backdoor attacks. More about graph backdoor attacks are in Appendix A.

**Graph Backdoor Defense.** UGBA (Dai et al., 2023) denotes that the attributes of triggers differ significantly from the attached poisoned nodes in GTA (Xi et al., 2021), thereby violating the homophily property typically observed in real-world graphs. Thus they propose a defense method called Prune, which removes edges that connect nodes with low similarity. DPGBA (Zhang et al., 2024) further indicates that although the triggers in UGBA (Dai et al., 2023) may demonstrate high similarity to target nodes, the triggers in both UGBA (Dai et al., 2023) and GTA (Xi et al., 2021) are still outliers. Thus they propose a defense method called OD, which involves training a graph auto-encoder and filtering out nodes with high reconstruction loss. More details about backdoor defense in non-structure data (Li et al., 2021a) and robust GNNs (Zhang & Zitnik, 2020; Zhu et al., 2019; Wang et al., 2021) are shown in Appendix A. Our work is inherently different from theirs: (i) We aim to defend against graph backdoor attacks regardless of the trigger attributes; (ii) we design a novel and theoretically guaranteed method to precisely identify poison nodes, aiming to degrade the attack success rate (ASR) without compromising clean accuracy.

## 3 PRELIMINARY

### 3.1 BACKGROUND AND PROBLEM DEFINITION

We use $\mathcal{G} = (\mathcal{V}, \mathcal{E}, \mathbf{X})$ to denote an attributed graph, where $\mathcal{V} = \{v_1, ..., v_N\}$ is the set of $N$ nodes, $\mathcal{E} \subseteq \mathcal{V} \times \mathcal{V}$ is the set of edges, and $\mathbf{X} = \{\mathbf{x}_1, ..., \mathbf{x}_N\}$ is the set of attributes of $\mathcal{V}$. $\mathbf{A} \in \mathbb{R}^{N \times N}$ is

the adjacency matrix of $\mathcal{G}$, where $\mathbf{A}_{ij} = 1$ if nodes $v_i$ and $v_j$ are connected; otherwise $\mathbf{A}_{ij} = 0$. In this paper, we focus on the inductive setting. Specifically, during the training stage, we are provided with a graph $\mathcal{G}_T = (\mathcal{V}_T, \mathcal{E}_T, \mathbf{X}_T)$. We use $\mathcal{V}_C \subseteq \mathcal{V}_T$ and $\mathcal{V}_B \subseteq \mathcal{V}_T$ to denote the clean node set and backdoored node set, respectively. A node $v_i \in \mathcal{V}_C$ is clean and labeled with the clean label $y_i$, whereas a node $v_j \in \mathcal{V}_B$ is backdoored and labeled with the target label $y_t$. The set $\mathcal{V}_T \setminus (\mathcal{V}_C \cup \mathcal{V}_B)$ denote the unlabeled nodes. The set of edges linking $v_i \in \mathcal{V}_B$ and the trigger $g_i$ is denoted as $\mathcal{E}_B \in \mathcal{E}_T$. During the inference stage, we are given an unseen graph $\mathcal{G}_U = (\mathcal{V}_U, \mathcal{E}_U, \mathbf{X}_U)$ where $\mathcal{V}_U = \mathcal{V}_{UC} \cup \mathcal{V}_{UB}$. A node $v_i \in \mathcal{V}_{UC}$ is clean, while a node $v_j \in \mathcal{V}_{UB}$ is backdoored. Notably, $\mathcal{G}_U$ is disjoint from the training graph $\mathcal{G}_T$, meaning $\mathcal{V}_U \cap \mathcal{V}_T = \emptyset$. The set of edges linking $v_j \in \mathcal{V}_{UB}$ and the trigger $g_j$ is denoted by $\mathcal{E}_{UB} \in \mathcal{E}_U$. The neighbors of node $v_i$ are denoted as $\mathcal{N}(i)$.

**Threat Model**. The attacker aims to add backdoor triggers, i.e., nodes or subgraphs, to a small set of target nodes $\mathcal{V}_B$ in the training graph and label them as a target class $y_t$, so that a GNN trained on the poisoned graph will (i) memorize backdoor triggers and be misguided to classify nodes attached with triggers as $y_t$; and (ii) behave normally for clean nodes without triggers attached. Specifically, given $\mathcal{V}_B \in \mathcal{V}$ as a set of target nodes, the attacker attaches the crafted triggers $g_i = (\mathbf{X}_i^g, \mathbf{A}_i^g)$ to the node $v_i \in \mathcal{V}_B$ and get poisoned node $\widetilde{v}_i = a(v_i, g_i)$, where $a(\cdot)$ is the attachment operation. Then, the attacker assigns $\mathcal{V}_B$ with target class $y_t$ to form backdoored graph $\mathcal{G}_T$.

**Defender's Knowledge and Capability**. During the training phase, the defender has access to the backdoored graph $\mathcal{G}_T$ to train a classifier for node classification. However, the defender lacks information about which nodes belong to the backdoor node set $\mathcal{V}_B$ and the target class $y_t$. During the inference phase, the defender is presented with an unseen backdoored graph $\mathcal{G}_U$ for node classification.

**Graph Backdoor Defense**. With the above description, our defense problem is formally defined as: *Given a backdoored graph $\mathcal{G}_T$, we aim to train a backdoor-free GNN model $f$ on $\mathcal{G}_T$ so that it can defend against backdoor triggers during inference on an unseen backdoored graph $\mathcal{G}_U$, while maintaining accuracy on clean data. This is formally defined as* $\min_f \sum_{v_i \in \mathcal{V}_{UC}} l(f(v_i), y_i) - \sum_{v_j \in \mathcal{V}_{UB}} l(f(\widetilde{v}_j), y_t)$, *where $l$ is the classification loss.*

## 3.2 PREDICTION VARIANCE CAUSED BY TRIGGER EDGE DROPPING

Generally, graph backdoor attacks establish an edge between a trigger and a target node (Xi et al., 2021; Dai et al., 2023; Zhang et al., 2021), which means the backdoor attack will fail if the edge linking the trigger and the target node is dropped. Furthermore, an explicit requirement for backdoor attacks is that the backdoored GNN model $f_b$ behaves normally for clean nodes without triggers attached. Based on the analysis above, we make an *assumption*: given a backdoored graph $\mathcal{G}_T$, if

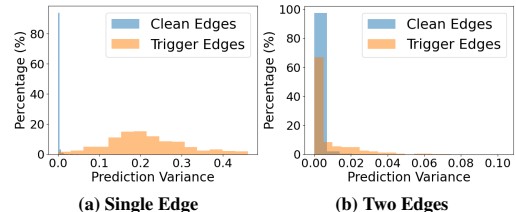

(a) Single Edge      (b) Two Edges

Figure 1: Prediction variance caused by dropping trigger edges and clean edges.

the trigger edge $e_i \in \mathcal{E}_B$ linking the target node $v_i \in \mathcal{V}_B$ and trigger $g_i$ is dropped, the prediction logits of the backdoored model for the poisoned target node $v_i$ will typically change significantly, compared to the prediction change on the nodes connected by clean edges $e_j \in \mathcal{E}_T \setminus \mathcal{E}_B$ when these edges $e_j$ are dropped.

To empirically verify our assumption, we conduct the following experiments: Given a backdoored graph, we first pretrain a backdoored model on this graph. For each node, we then iteratively drop each of its neighbors one by one and measure the prediction variance caused by dropping trigger edges $\mathcal{E}_B$ and clean edges $\mathcal{E}_T \setminus \mathcal{E}_B$, respectively. We perform experiments on OGB-arxiv (Hu et al., 2020) with 565 triggers. The attack method is DPGBA (Zhang et al., 2024). The model architecture is a 2-layer GCN (Kipf & Welling, 2016). From the results shown in Fig. 1 (a), it is evident that dropping adversarial edge connecting backdoor trigger will result in a much larger prediction variance than dropping clean edges. More results on other datasets can be found in Appendix B.

Thus, an intuitive method for identifying a backdoor trigger or poisoned target node is to examine each edge individually. For instance, in a 2-layer GCN (Kipf & Welling, 2016), we remove one of its neighbors at a time for each node and conduct inference based on the remaining 2-hop neighbors. By observing prediction changes with each neighbor removal, we estimate the likelihood of an edge being linked to the backdoor trigger: greater prediction change indicates a higher probability.

However, this method has several issues: (i) *Scalability*: For graphs with high average degree, this method is computationally expensive. Specifically, for an $L$-layer GCN, the time complexity is $\mathcal{O}(LNd^L M(d+M))$, where $d$ is the average degree, $N$ is the number of nodes, and $M$ denotes the feature dimension. The analysis is in Appendix C. As $d$ grows, the time complexity grows exponentially with the power $L$, making it impractical for dense graphs. (ii) *Utility*: When the backdoor trigger is a subgraph with multiple edges linking it to a target node, examining each edge individually becomes ineffective. As shown in Fig. 1 (b), when two edges linking a trigger and a target node, on the OGB-arxiv dataset, most of the prediction variance caused by dropping trigger edges shows similar values to those caused by dropping clean edges, rendering the intuitive method ineffective.

Therefore, while prediction variance caused by edge dropping is a crucial indicator for identifying poisoned nodes or backdoor triggers, it is essential to develop a method that can (i) accurately identify poisoned nodes, even when multiple edges connect a trigger to a target node; and (ii) reduce the time complexity to scale linearly with the average degree, thereby improving scalability.

**Vulnerability of Clean Model to Backdoor Triggers.** Even if we can remove backdoor triggers from the training dataset, a graph backdoor trigger can still potentially lead to a successful attack as the attacker can craft a backdoor trigger that mimics the neighbor of nodes (Zhang et al., 2024) from the target class. To verify this, we adopt the DPGBA (Zhang et al., 2024) attack method and conduct experiments by removing the backdoor triggers from the training dataset. We then train a 2-layer GCN on this clean dataset and report the attack success rate (ASR) on the test dataset, as shown in Table 1. The Random ASR is calculated by $\frac{1}{C}$ as a reference, where $C$ is the number of classes for each dataset. As shown in the table, even when the model is trained on a clean dataset, it can still react to a backdoor trigger that mimics the neighbor of nodes from the target class. Thus, simply removing the backdoor triggers from the training dataset is insufficient to defend against graph backdoor attacks.

Table 1: ASR on clean model.

|  | Cora | OGB-arxiv |
|---|---|---|
| Rand. ASR | 14.29 | 2.5 |
| ASR | **34.72** | **5.4** |

## 4 METHODOLOGY

Our preliminary analysis shows that (1) removing adversarial edges linking triggers and poisoned target nodes typically results in higher prediction variance for poisoned target nodes; (2) simply removing backdoor triggers from the training dataset is insufficient to protect against backdoor triggers that mimic the distribution of neighboring nodes from the target class. To utilize the above information and solve our problem, we face two challenges: (1) How to devise an efficient method to find poisoned target nodes? (2) How to minimize the impact of back-

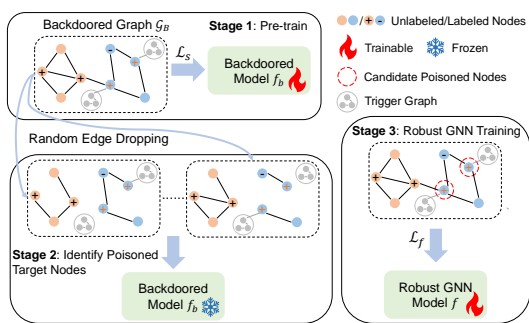

Figure 2: Framework of RIGBD

door triggers without degrading the clean accuracy? To address the above challenges, a novel framework RIGBD is proposed. Specifically, to address the first challenge, we theoretically and empirically verify that random edge dropping is an efficient way to distinguish the poisoned target nodes from clean nodes. To address the second challenge, we propose to minimize the prediction confidence of poisoned target nodes on the target class. This encourages the model to counteract the impact of the trigger. Next, we give the details of each component.

### 4.1 IDENTIFYING POISONED TARGET NODES BY RANDOM EDGE DROPPING

To overcome the scalability and utility issues of the intuitive method of dropping each edge individually, we propose a novel random edge dropping framework. Specifically, we define a randomized edge drop noise $\epsilon$ for adjacency matrix $\mathbf{A}$ that removes each existing edge with the probability $\beta$ as

$$P(\epsilon_{ij} = 0 \mid \mathbf{A}_{ij} = 0) = 1, P(\epsilon_{ij} = 1 \mid \mathbf{A}_{ij} = 1) = \beta, P(\epsilon_{ij} = 0 \mid \mathbf{A}_{ij} = 1) = 1 - \beta, \quad (1)$$

where $\epsilon_{ij} = 1$ means dropping edge $e_{ij}$. We denote $\mathbf{A} \oplus \epsilon$ as the perturbed graph, with $\oplus$ the element-wise XOR operator. Given a backdoored graph $\mathcal{G}_T$ and corresponding adjacency matrix

$\mathbf{A}$, we first train the backdoored node classifier $f_b$ on this poisoned graph $\mathcal{G}_T$ and get the prediction logits $\mathbf{p}_i$ for each node $v_i \in \mathcal{V}_L$, i.e., $\mathbf{p}_i = f_b(\mathbf{A}; v_i)$. Then, we independently add random noise $\epsilon$ to the original adjacency matrix $\mathbf{A}$ for $K$ times using the operation $\mathbf{A} \oplus \epsilon$, resulting in the noisy matrices $\mathbf{A}_1, \ldots, \mathbf{A}_K$. Similarly, we obtain the prediction logits $\mathbf{p}_i^k = f_b(\mathbf{A}_k; v_i)$ for node $v_i$ on each perturbed graph. Then, the prediction variance of node $v_i$ against random edge dropping is:

$$s(i) = \sum\nolimits_{k=1}^{K} \mathrm{KL}(\mathbf{p}_i, \mathbf{p}_i^k), \tag{2}$$

where $\mathrm{KL}(\mathbf{p}_i, \mathbf{p}_i^k)$ denotes Kullback-Leibler divergence between $\mathbf{p}_i$ and $\mathbf{p}_i^k$. With an appropriate $K$ and $\beta$, it is highly likely that an edge connecting the backdoor trigger will be removed in one or more of the perturbed adjacency matrices, causing significant prediction variance for poisoned nodes.

However, it is unclear whether this method can also lead to large prediction variance for clean nodes, potentially compromising our ability to differentiate between poisoned target nodes and clean ones. To address this concern, we propose to obtain each node representation only based on its neighbors. Specifically, when performing inference to obtain the prediction logits $\mathbf{p}_i$ and $\mathbf{p}_i^k$ (before and after random edge dropping), the operation in each layer can be written as:

$$\mathbf{H}^{(l+1)} = \sigma \left( \tilde{\mathbf{D}}^{-\frac{1}{2}} \mathbf{A} \tilde{\mathbf{D}}^{-\frac{1}{2}} \mathbf{H}^{(l)} \mathbf{W}^{(l)} \right). \tag{3}$$

where $\mathbf{A}$ is the adjacency matrix without self-connections. In contrast, GCN (Kipf & Welling, 2016) obtain node representations by $\mathbf{H}^{(l+1)} = \sigma(\tilde{\mathbf{D}}^{-\frac{1}{2}}(\mathbf{A} + \mathbf{I})\tilde{\mathbf{D}}^{-\frac{1}{2}} \mathbf{H}^{(l)} \mathbf{W}^{(l)})$ where $\mathbf{I}$ is the identity matrix.

We adopt the graph convolution strategy in Eq. 3 to get node representations for the following reasons: (1) For clean nodes, as each neighbor has the same drop ratio $\beta$, in expectation, the node representation is unchanged because the proportion of the expected contribution of each neighbor's features to the node representation is maintained. Thus, the expectation of prediction variance for clean nodes tends to be small. In contrast, if we include the attributes of the central node, the expectation of the node representation will focus more on the central node after random edge dropping, making the prediction variance unpredictable. We will analyze further in Theorem 1. (2) For poisoned target node, as backdoor triggers are designed to attack various types of targets, once the trigger exists after random edge dropping, it will still lead to a successful attack, regardless of whether the node representations are obtained only based on neighbors or also consider the node itself. However, if the trigger is dropped, the model will exhibit a large prediction variance. In cases where all the neighbors of a central node are dropped, we cancel the edge drop operation for that node and retain its original neighbors. Notably, while the attributes of the central node are important for accurate classification, our focus here is on testing the prediction variance and identifying poisoned nodes. Once target nodes are identified, various GNNs can be used to train a classifier with our robust loss in Eq. 6.

To prove that our method of randomized edge dropping can effectively distinguish poisoned target nodes from clean nodes, we make the following assumptions on graph and propose theorems.

**Assumptions on Graphs.** Following (Dai et al., 2022b; Ma et al., 2021a), we consider a graph $\mathcal{G}$, where each node $v_i$ has features $\mathbf{x}_i \in \mathbb{R}^m$, a label $y_i$ and $\deg(i)$ denotes the number of its neighbors. We assume that: (1) The feature of node $v_i$ is sampled from the feature distribution $F_{y_i}$ that depends on the label $y_i$ of the node, i.e., $\mathbf{x}_i \sim \mathcal{F}_{y_i}$. (2) Feature dimensions of $\mathbf{x}_i$ are independent to each other; (3) The features in $\mathbf{X}$ are bounded by a positive scalar $B$, i.e., $\max_{i,j} |\mathbf{X}[i, j]| \leq B$.

**Theorem 1.** *Consider a graph $\mathcal{G} = \{\mathcal{V}, \mathcal{E}, \mathbf{X}\}$ following Assumptions (1)-(3). For clean node $v_i \in \mathcal{V}$ and its neighbors $\mathcal{N}(i)$, the expectation of the pre-activation output of a single operation defined in Eq. 3 is given by $\mathbb{E}[\mathbf{h}_i]$. Then the expectation of the pre-activation output of a single operation defined in Eq. 3 after random edge dropping is given by $\mathbb{E}[\mathbf{h}_i^k] = \mathbb{E}[\mathbf{h}_i]$.*

The proof is in Appendix D.1. Theorem 1 shows that, in expectation, the output embedding of a node remains unchanged before and after random edge dropping when applying the layer operation defined in Eq. 3, regardless of the drop ratio. However, in practice, conducting numerous experiments is not feasible. Thus, we analyze the distance between the observed node embeddings after random edge dropping and the expectation of node embeddings without perturbation.

**Theorem 2.** *Consider a graph $\mathcal{G} = \{\mathcal{V}, \mathcal{E}, \mathbf{X}\}$ following Assumptions (1)-(3). Let $\mathbf{h}_i$ and $\mathbf{h}_i^k$ denote the clean node embedding before and after the $k$-th random edge dropping, respectively, and*

$\deg(i)_k$ *represent the corresponding degree of node $v_i$. The probability that the distance between the observation $\mathbf{h}_i^k$ and the expectation of $\mathbf{h}_i$ is larger than $t$ is bounded by:*

$$\mathbb{P}\left(\left\|\mathbf{h}_i^k - \mathbb{E}\left[\mathbf{h}_i\right]\right\|_2 \geq t\right) \leq 2 \cdot M \cdot \exp\left(-\frac{\deg(i)_k t^2}{2\rho^2(\mathbf{W})B^2 M}\right), \tag{4}$$

*where $M$ denotes the feature dimension and $\rho^2(\mathbf{W})$ denotes the largest singular value of $\mathbf{W}$.*

The proof is in Appendix D.2. Theorem 2 shows that the distance between the observed node embeddings after random edge dropping and the expectation of node embeddings without perturbation is small with a high probability. Furthermore, nodes with higher degrees are more likely to exhibit less variation in their embeddings. This highlights the effectiveness of our method of random edge dropping, particularly in dense graphs, compared to the method that drops each edge individually.

Based on the above analysis, we theoretically demonstrate that our method, which involves random edge dropping and obtaining node representations through graph convolution as defined in Eq. 3, results in small changes to node representations. Thus, for **clean nodes**, this stability ensures that the prediction output is consistent as the backdoored model does not exhibit sensitivity towards any particular clean neighbors. However, for **poisoned target nodes**, the scenario differs. Despite the stability in their node representations after random edge dropping, the last-layer classifier's sensitivity (Chen et al., 2018) to the presence or absence of the backdoor trigger leads to a significant variance in prediction outcomes compared to clean nodes. This is verified in Sec. 3.2.

Next, we analyze the expected number of times the backdoor trigger $g_i$ can be removed from the neighbor of a poisoned node $v_i$ after $K$ iterations of random edge dropping. This also indicates the expected number of times the poisoned node $v_i$ demonstrates a larger prediction variance.

**Theorem 3.** *Without loss of generality, we consider the case where there is only one edge connecting a backdoor trigger to a target node. Let $\beta$ denote the random edge dropping ratio, and $K$ be the total number of iterations for random edge dropping and conducting inference on the perturbed graph. For a poisoned node $v_i$ and a trigger $g_i$, the expected number of times the poisoned node $v_i$ demonstrates large prediction variance is given as $\mathbb{E}(g_i)_d = K \cdot \beta$.*

Theorem 3 demonstrates that this expected number grows fast with a certain drop ratio. Since Theorem 1 indicates that, in expectation, the output embedding of a clean node remains unchanged before and after perturbation, regardless of the drop ratio, we can use a high drop ratio, e.g. $\beta = 0.5$, to achieve a fast increase in the expected number of times the poisoned node $v_i$ demonstrates large prediction variance. Our experimental results in Section 5.4 show that with $\beta = 0.5$, even a small value of $K$, e.g. $K = 5$, can already identify most of the poisoned target nodes from the clean ones.

To empirically verify that our method of random edge dropping can efficiently distinguish poisoned nodes from clean nodes, we adopt the attack method DPGBA and conduct experiments on Cora and OGB-arxiv to show the prediction variance of poisoned nodes and clean nodes after random edge dropping. The details of the experiment settings and the results are in Appendix E. From the results, we observe that: **(i)** Our method consistently results in higher prediction variance for poisoned nodes, enabling the distinction between poisoned and clean nodes. **(ii)** Even when two edges link a backdoor trigger to a poisoned node, our method maintains high prediction variance for most poisoned nodes. This showcases the superior performance of our method compared to the intuitive approach of dropping each edge individually. Furthermore, the time complexity of our method scales linearly with $K$, which is typically small, whereas the intuitive method scales exponentially with $d^L$, leading to significantly higher computational costs as $d$ increases. We also compare the prediction variance of low-degree nodes from both homophilic and heterophilic graphs against poisoned nodes in Appendix F.

**Identify Target Nodes and Target Class**. Next, we give details of how we determine the target class $y_t$ and identify the candidate poisoned target node set $\mathcal{V}_s$. Given $s(i)$ as the prediction variance for each node $v_i \in \mathcal{V}_T$ with label $y_i$, we sort nodes by $s(i)$ in descending order and form the set $\mathcal{D}' = \{(s_{\sigma(1)}, y_{\sigma(1)}), (s_{\sigma(2)}, y_{\sigma(2)}), \ldots, (s_{\sigma(n)}, y_{\sigma(n)})\}$, where $\sigma$ is a permutation of $\{1, 2, \ldots, n\}$ such that the sorted values satisfy $s_\sigma(1) > s_\sigma(2) > \cdots > s_\sigma(n)$. Then we determine the target class as the label of the node with the largest prediction variance, i.e., $y_t = y_{\sigma(1)}$. Let $j$ be the index of the first entry in $\mathcal{D}'$ such that $y_{\sigma(j)} \neq y_t$ and $y_{\sigma(j+1)} \neq y_t$. The threshold of prediction variance to select candidate poisoned target nodes is defined as:

$$\tau = s_{\sigma(j)} \quad \text{where} \quad j = \min\left\{k \mid y_{\sigma(k)} \neq y_t \text{ and } y_{\sigma(k+1)} \neq y_t,, k = 1, 2, 3, \ldots, n\right\}. \tag{5}$$

Then, we select nodes with prediction variance larger than the threshold $\tau$ as candidates for poisoned target nodes, denoted as $\mathcal{V}_s$.

## 4.2 BACKDOOR ROBUST GNN MODEL TRAINING

Though random edge dropping can help identify the most poisoned target nodes, there are still some concerns. *First*, a small amount of poisoned target nodes might exhibit prediction variances similar to those of clean nodes, inevitably remaining within the graph. *Second*, as discussed in Section 3.2, merely eliminating backdoor triggers from the training set is insufficient to defend against backdoor triggers that mimic the neighbor distribution of nodes from the target class.

An intuitive solution is to train a trigger detector capable of distinguishing clean nodes and backdoor triggers. Then, during inference on an unseen graph $\mathcal{G}_U$, the trigger detector is adopted to remove potential triggers. However, when the trigger is a subgraph, multiple nodes, and their interactions need to be considered simultaneously during training, complicating the process of training the trigger detector. Additionally, each time we are given an unseen graph, we need to run the trigger generator to perform inference on the entire graph first, increasing the computational cost.

Thus, in our RIGBD, we propose directly training a backdoor robust GNN node classifier $f$ on the training dataset $\mathcal{V}_L$ by minimizing its prediction confidence on the target class $y_t$ for poisoned nodes, thereby encouraging the model to counteract the impact of the backdoor triggers and be robust against backdoor attack. Specifically, given the selected poisoned target nodes $\mathcal{V}_s \in \mathcal{V}_T$, the objective function to train a backdoor robust GNN node classifier $f$ is:

$$\min_f \mathcal{L}_f = \sum_{v_i \in \mathcal{V}_s} \log f(v_i)_{y_t} + \sum_{v_j \in \mathcal{V}_L \backslash \mathcal{V}_s} \mathcal{L}(f(v_j), y_j), \qquad (6)$$

where $f(v_i)_{y_t}$ denotes the prediction confidence of $f$ for $v_i$ on target class $y_t$ and $\mathcal{L}(f(v_j), y_j)$ is the cross entropy loss. The resulting classifier is robust against triggers as (i) We explicitly countermeasure the detected backdoors, making our model robust to backdoor. Though there might be target nodes not detected, they generally have smaller prediction variance, meaning that they have less impact on the training of $f$. As these triggers more or less have patterns similar to the detected ones, our training strategy implicitly mitigates their effects. (ii) For triggers that mimic the neighbor distribution of nodes from the target class, our model $f$ is encouraged to explore the subtle differences between clean nodes from the target class and poisoned target nodes, thereby ensuring its clean accuracy. The training algorithm is in Appendix G.

## 5 EXPERIMENTS

In this section, we conduct experiments to answer the following research questions: **(Q1)** How effective is RIGBD in defending against graph backdoor attacks? **(Q2)** How is the performance of RIGBD in detecting poisoned nodes? **(Q3)** How do different drop ratios and different numbers of iterations of random edge dropping impact the performance of RIGBD?

### 5.1 EXPERIMENTAL SETUP

**Datasets.** We conduct experiments on three benchmark datasets widely used for node classification, i.e., Cora, Citeseer, Pubmed (Sen et al., 2008), Physics (Sinha et al., 2015), Flickr (Zeng et al., 2019) and OGB-arxiv (Hu et al., 2020). More details of the datasets are summarized in Appendix H.1.
**Attack Methods.** To demonstrate the defense ability of our RIGBD, we evaluate RIGBD on 3 state-of-the-art graph backdoor attack methods, i.e. GTA (Xi et al., 2021), UGBA (Dai et al., 2023) and DPGBA (Zhang et al., 2024). The details of these attacks are given in Appendix H.2.
**Compared Methods.** We implement the backdoor defense strategies Prune (Dai et al., 2023) and OD (Zhang et al., 2024). Additionally, three representative robust GNNs, i.e. RobustGCN (Zhu et al., 2019), GNNGuard (Zhang & Zitnik, 2020) and randomized smoothing (RS) (Wang et al., 2021) with various drop ratios are also selected. We also include ABL (Li et al., 2021a), which is a popular backdoor defense method in the image domain and aims to train clean models given backdoor-poisoned data. More details of these defense methods are given in Appendix H.3.
**Evaluation Protocol.** Following existing representative graph backdoor attacks (Dai et al., 2023; Zhang et al., 2024), we split the graph into two disjoint subgraphs, $\mathcal{G}_T$ and $\mathcal{G}_U$, with an $80 : 20$ ratio.

Table 2: Results of backdoor defense.

| Attacks | Defense | Cora | | Citeseer | | PubMed | | Physics | | Flickr | | OGB-arxiv | |
|---|---|---|---|---|---|---|---|---|---|---|---|---|---|
| | | ASR(%)↓ | ACC(%)↑ | ASR(%)↓ | ACC(%)↑ | ASR(%)↓ | ACC(%)↑ | ASR(%)↓ | ACC(%)↑ | ASR(%)↓ | ACC(%)↑ | ASR(%)↓ | ACC(%)↑ |
| GTA | GCN | 98.98 | 82.58 | 100.00 | 73.70 | 93.09 | 85.18 | 96.42 | 89.30 | 88.45 | 41.23 | 75.34 | 65.76 |
| | GNNGuard | 40.22 | 78.52 | 55.26 | 63.55 | 26.93 | 81.68 | 43.87 | 78.72 | 0.00 | 40.40 | 0.04 | 62.58 |
| | RobustGCN | 90.46 | 80.37 | 95.31 | 73.79 | 93.12 | 81.68 | 95.47 | 94.84 | 85.42 | 42.26 | 70.95 | 56.08 |
| | Prune | 17.63 | 83.06 | 12.24 | 72.46 | 28.10 | 85.05 | 8.34 | 88.45 | 12.56 | 41.27 | 0.01 | 63.97 |
| | OD | 0.04 | 83.47 | 0.04 | 72.84 | 0.03 | 85.27 | 0.12 | 90.24 | 2.12 | 41.42 | 0.01 | 65.23 |
| | RS | 53.14 | 73.33 | 52.86 | 65.66 | 42.28 | 84.58 | 57.70 | 92.52 | 52.85 | 42.31 | 42.72 | 58.48 |
| | ABL | 19.93 | 81.85 | 13.09 | 73.19 | 16.18 | 83.92 | 16.82 | 92.06 | 17.54 | 41.46 | 11.28 | 63.24 |
| | RIGBD | 0.00 | 83.70 | 0.34 | 74.10 | 0.01 | 85.13 | 0.32 | 95.10 | 0.00 | 44.21 | 0.01 | 66.51 |
| UGBA | GCN | 98.76 | 83.42 | 100.00 | 74.70 | 96.42 | 84.64 | 100.00 | 95.94 | 93.14 | 44.71 | 98.82 | 63.95 |
| | GNNGuard | 43.17 | 78.15 | 94.53 | 64.76 | 98.97 | 81.48 | 95.26 | 87.54 | 96.93 | 42.15 | 95.21 | 64.61 |
| | RobustGCN | 98.67 | 80.00 | 99.82 | 71.69 | 99.90 | 82.85 | 99.94 | 94.08 | 90.35 | 41.82 | 90.35 | 56.18 |
| | Prune | 98.89 | 82.66 | 97.68 | 74.35 | 92.87 | 85.09 | 94.67 | 93.87 | 91.43 | 43.65 | 93.07 | 62.58 |
| | OD | 0.03 | 83.65 | 0.06 | 73.80 | 0.01 | 85.19 | 0.02 | 95.36 | 1.65 | 43.57 | 0.01 | 65.35 |
| | RS | 54.24 | 70.37 | 50.34 | 69.88 | 44.41 | 84.68 | 47.77 | 94.29 | 20.69 | 42.18 | 40.30 | 58.76 |
| | ABL | 15.13 | 81.48 | 12.08 | 73.19 | 28.60 | 84.37 | 14.87 | 94.69 | 15.32 | 41.66 | 34.26 | 64.93 |
| | RIGBD | 0.01 | 84.81 | 0.00 | 73.80 | 0.01 | 85.13 | 0.12 | 95.71 | 0.00 | 43.66 | 0.01 | 65.21 |
| DPGBA | GCN | 97.72 | 83.34 | 100.00 | 74.09 | 98.63 | 85.22 | 100.00 | 95.59 | 90.79 | 44.87 | 95.63 | 65.72 |
| | GNNGuard | 85.61 | 78.52 | 46.98 | 60.84 | 44.12 | 80.82 | 88.72 | 88.76 | 95.85 | 43.52 | 94.66 | 62.29 |
| | RobustGCN | 96.68 | 81.11 | 91.64 | 71.08 | 94.88 | 82.54 | 91.25 | 94.87 | 24.60 | 41.69 | 90.09 | 60.38 |
| | Prune | 91.82 | 85.28 | 94.80 | 73.21 | 88.64 | 85.13 | 94.27 | 94.73 | 88.96 | 44.75 | 90.47 | 65.53 |
| | OD | 94.33 | 83.58 | 98.42 | 73.66 | 91.32 | 85.12 | 98.72 | 95.48 | 90.42 | 43.63 | 93.30 | 65.47 |
| | RS | 51.29 | 69.63 | 50.34 | 71.08 | 48.83 | 85.44 | 48.19 | 95.30 | 27.94 | 42.21 | 41.18 | 58.44 |
| | ABL | 86.72 | 79.26 | 11.41 | 73.49 | 48.58 | 79.45 | 11.74 | 95.30 | 30.42 | 41.71 | 52.56 | 63.88 |
| | RIGBD | 0.01 | 85.19 | 0.33 | 73.79 | 0.03 | 84.56 | 0.21 | 95.79 | 0.00 | 43.78 | 0.00 | 65.24 |

The graph $\mathcal{G}_T$ is used to train the attacker. Then, the attacker selects target nodes $\mathcal{V}_B$ and attaches triggers to these target nodes to form the backdoored graph $\mathcal{G}_T$. The number of triggers $|\mathcal{V}_B|$ is set as 40, 80, 160, 160, 160 and 565 for Cora, Citeseer, PubMed, Physics, Flickr, and OGB-arxiv, respectively. The trigger size is limited to three nodes in all experiments. The defender trains a model on poisoned graph $\mathcal{G}_T$. Next, half of the nodes in $\mathcal{G}_U$ are selected as poisoned nodes and are attached with backdoor triggers to test the attack success rate (ASR). The remaining nodes in $\mathcal{G}_U$ are kept clean and used to test the clean accuracy (ACC). We also report the Recall and Precision of our method in identifying poisoned nodes. Recall is defined as the percentage of poisoned nodes among the candidate nodes identified, relative to all poisoned nodes. Precision is defined as the percentage of poisoned nodes among the candidate nodes identified. Our RIGBD deploys a 2-layer GCN as the model architecture. Each experiment is conducted 5 times and the average results are reported.

## 5.2 PERFORMANCE OF DEFENSE

To answer **Q1**, we compare RIGBD with the baseline defense methods across three datasets. The number of iterations for random edge dropping is set to $K = 20$, with a drop ratio of $\beta = 0.5$. We report ASR and ACC in Table 2. From the table, we observe: **(i)** Across all datasets and attack methods, RIGBD consistently achieves the lowest ASR score, typically close to $0\%$. Although defense methods such as Prune and OD perform well against GTA and UGBA attacks, they fail to defend against DPGBA, which generates in-distribution triggers. This indicates that RIGBD is highly effective in defending against various types of backdoor triggers and backdoor attacks. **(ii)** Our RIGBD achieves comparable or slightly better clean accuracy compared to the vanilla GCN. This indicates that our approach of random edge dropping typically leads to higher prediction variance for poisoned nodes, effectively identifying backdoor triggers. In cooperation with this precise identification of poisoned nodes, our strategy to train a backdoor robust model using Eq. 6 can significantly reduce the ASR while maintaining clean accuracy. Additional results on RS with varying drop ratios and RIGBD using different GNN architectures can be found in Appendix J.

## 5.3 ABILITY TO DETECT POISONED NODES

To answer **Q2**, we present the recall and precision of RIGBD in identifying poisoned nodes. Using the same setting as in Sec. 5.2, the results on OGB-arxiv are shown in Table 3. We also include

Table 3: Results for the ability to detect poisoned nodes.

| Attacks | Clean ACC | ASR | ACC | Recall | Precision |
|---|---|---|---|---|---|
| GTA | 65.8 | 0.01 | 66.51 | 84.9 | 90.3 |
| UGBA | 65.8 | 0.01 | 65.21 | 95.6 | 96.3 |
| DPGBA | 65.8 | 0.00 | 65.24 | 90.5 | 93.9 |

the corresponding ASR and ACC to illustrate the correlation between detection ability and defense performance. Clean accuracy (Clean ACC), obtained from a model trained on the clean graph, is provided as a reference to evaluate the model's performance on clean nodes. From the table, we observe the following: **(i)** Our RIGBD demonstrates consistently high precision, always over 90%, in detecting poisoned nodes across three attack methods, with detection recall always exceeding 80%. This indicates that our strategy of random edge dropping typically leads to higher prediction variance

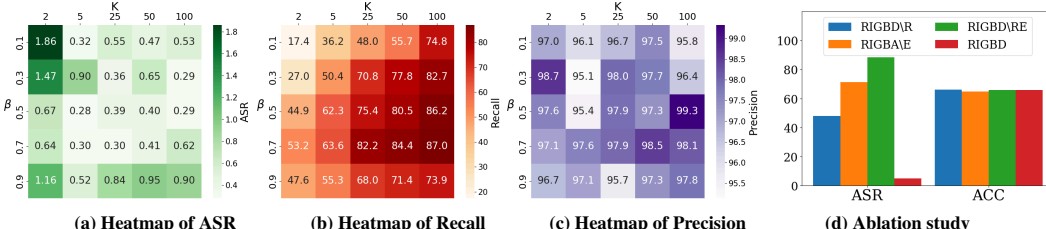

Figure 3: Hyperparameter Sensitivity Analysis and Ablation Study.

for poisoned nodes. **(ii)** Although the recall for the GTA attack method is 84.9%, we still achieve an ASR close to 0% while maintaining ACC. This verifies the stability of our strategy to train a robust node classifier using Eq. 6. Even when some poisoned nodes are not identified, it still efficiently helps the model unlearn the triggers. More results on other datasets are in Appendix L.

## 5.4 HYPERPARAMETER ANALYSIS AND ABLATION STUDY

**Hyperparameter Analysis**. To answer **Q3**, we conduct experiments to show how different drop ratios $\beta$ and different numbers of iterations of random edge dropping $K$ impact the performance of RIGBD. Specifically, we vary the value of $K$ as $\{2, 5, 25, 50, 100\}$, and the values of $\beta$ as $\{0.1, 0.3, 0.5, 0.7, 0.9\}$. The attack method used is DPGBA. The other settings are the same as the evaluation protocol in Sec. 5.1. The results on OGB-arxiv are shown in Fig. 3. From the figure, we observe: **(i)** Our RIGBD is stable in terms of ASR and Precision. Notably, when the drop ratio $\beta = 0.1$ and $K = 2$, the recall of poisoned node detection is around 17.4%. However, we still achieve robust defense performance with 1.86% ASR. This is because our framework identifies the most efficient triggers that lead to large prediction variance and focuses on minimizing their impact. By doing so, the influence of less efficient triggers is inherently counteracted. In contrast, when we simply remove those 17.4% triggers from the dataset and train a normal GNN on the remaining dataset, the ASR is around 80%. **(ii)** As $K$ increases, the recall also increases, empirically verifying our theorems that the clean nodes will have stable node embeddings and predictions. As $\beta$ increases from 0.1 to 0.7, the recall also increases without a decrease in precision. When $\beta = 0.9$, only about 10% of edges remain in each iteration. Although the recall decreases slightly, we still achieve consistently high precision. This further shows the robustness of RIGBD against various chosen hyperparameters. More results on the hyperparameter analysis for ACC are in Appendix M.

**Ablation Study**. To evaluate the effectiveness of random edge dropping, we replace it with the intuitive method of dropping edges individually, as described in Section 3.2, and obtain a variant named as RIGBD\E. We then conduct experiments focusing on scenarios where a backdoor trigger (subgraph) is linked to a poisoned node by two edges. To demonstrate the effectiveness of our robust training strategy, we implement a variant of our model, named RIGBD\R, which simply removes identified candidate poisoned nodes from the dataset and trains a GNN classifier with cross-entropy. We also implement a variant called RIGBD\RE, which involves individually dropping each edge and eliminating candidate poisoned nodes. The number of iterations for random edge dropping is set to $K = 20$, with a drop ratio of $\beta = 0.5$. The attack method used is DPGBA. The dataset is OGB-arxiv. We report the ASR and ACC in Fig. 3. From the figure, we observe: **(i)** When two edges link a backdoor trigger to a target node, the intuitive method of dropping each edge individually fails to defend against this attack. Notably, RIGBD\R achieves an ASR almost 40% lower than RIGBD\RE, highlighting that our random edge dropping is more efficient than dropping edges individually in identifying poisoned nodes. **(ii)** In cooperation with a robust training strategy, our RIGBD further achieves an ASR close to 0%. This underscores the superiority of our method in defending against various types of backdoor attacks. Additional results on the ablation study for the case where one edge links a trigger and a target node can be found in Appendix N.

**Impact of Various Numbers of Triggers and Trigger Sizes**. The performance and analysis of RIGBD against various attack budgets and trigger sizes are provided in Appendix O and P.

## 6 CONCLUSION

In this paper, we first empirically show that poisoned nodes exhibit high prediction variance with edge dropping. We then propose random edge dropping, supported by theoretical analysis, to efficiently

and precisely identify poisoned nodes. Additionally, we introduce a robust training strategy to develop a backdoor-robust GNN model, even if some poisoned nodes remain unidentified. Extensive experiments demonstrate that our method accurately detects poisoned nodes, significantly reduces the attack success rate, and maintains clean accuracy against various graph backdoor attacks.

## ACKNOWLEDGMENT

This material is based upon work supported by, or in part by the Army Research Office (ARO) under grant number W911NF-21-10198, the Department of Homeland Security (DHS) under grant number 17STCIN00001-05-00, and Cisco Faculty Research Award. The views and conclusions contained in this material are those of the authors and should not be interpreted as necessarily representing the official policies, either expressed or implied, of the funding agencies.

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

## A  DETAILS OF RELATED WORKS

### A.1  GRAPH BACKDOOR ATTACKS

Backdoor attacks have been extensively studied in the image domain (Li et al., 2022; Gu et al., 2019; Chen et al., 2017; Sun et al., 2023; Li et al., 2023). Initial works focused on directly poisoning training samples (Liu et al., 2020; Chen et al., 2017; Xing et al., 2024), while others explored making the triggers invisible (Li et al., 2021b; Doan et al., 2021). Additionally, hidden backdoors can be embedded through transfer learning (Kurita et al., 2020), modifying model parameters (Chen et al., 2021), and adding extra malicious modules (Tang et al., 2020). Recently, studies have begun investigating backdoor attacks on GNNs (Dai et al., 2023; Xi et al., 2021; Zhang et al., 2021; 2024; Lin et al., 2024), which differ from the more common poisoning and evasion attacks (Dai et al., 2024; 2022a; Ma et al., 2021b). Backdoor attacks involve injecting malicious triggers into the training data, causing the model to make incorrect predictions when these triggers are present in test samples. This type of attack subtly manipulates the training phase, ensuring the model performs as expected under normal conditions but fails when trigger-embedded inputs are present. Among pioneering efforts, SBA (Zhang et al., 2021) introduced a method for injecting universal triggers into training samples using a subgraph-based approach, though its attack success rate was low. GTA (Xi et al., 2021) advanced this by developing a technique for generating adaptive triggers, customizing perturbations for individual samples to enhance attack effectiveness. In UGBA (Dai et al., 2023), an algorithm for selecting poisoned nodes was introduced to optimize the attack budget, along with an adaptive trigger generator to create triggers with high cosine similarity to the target node. DPGBA (Zhang et al., 2024) demonstrate that existing backdoor attack methods on graphs suffer from either a low attack success rate or outlier issues. To address these problems, they propose an adversarial learning strategy to generate in-distribution triggers and introduce a novel loss function to enhance the attack success rate with these in-distribution triggers. In this paper, we investigate defense mechanisms against backdoor attacks that involve attaching backdoor triggers to target nodes (Xi et al., 2021; Dai et al., 2023; Zhang et al., 2024). These attacks are considered a substantial threat to real-world applications of GNNs, especially in comparison to attacks involving direct manipulation of node attributes (Xing et al., 2024). In real-world contexts, such as social media networks, it is more practical for an adversary to introduce malicious nodes (e.g., fake accounts) and connect them to a target node rather than altering the target node's attributes. Thus, it is anticipated that this form of attack will receive increasing research attention. Given this threat, it is crucial to develop robust defense frameworks capable of mitigating these types of attacks.

### A.2  GRAPH BACKDOOR DEFENSE

Backdoor defenses have been extensively investigated in the context of non-structured data. To address the threat of backdoor attacks, various defenses have been proposed, which can be broadly classified into two main categories: empirical backdoor defenses and certified backdoor defenses. Empirical backdoor defenses (Wang et al., 2019; Kolouri et al., 2020; Li et al., 2021a) are based on specific observations or understandings of existing attacks and generally perform well in practice; however, they lack theoretical guarantees and may be circumvented by adaptive attacks. On the other hand, certified backdoor defenses (Wang et al., 2020; Weber et al., 2023; Xie et al., 2021) offer theoretical guarantees of their validity under certain assumptions, but their practical performance is often weaker than empirical defenses due to the difficulty in meeting these assumptions. Improving defenses against backdoor attacks remains a significant open challenge. Limited defenses have been proposed against the graph backdoor. UGBA (Dai et al., 2023) highlights that in attack methods GTA (Xi et al., 2021), the attributes of triggers differ significantly from the attached poisoned nodes, violating the homophily property typically observed in real-world graphs. To address this, they propose a defense method called Prune, which involves removing edges that connect nodes with low cosine similarity. This approach significantly reduces ASR of previous works. DPGBA (Zhang et al., 2024) further notes that although the generated triggers in UGBA (Dai et al., 2023) may demonstrate high cosine similarity to target nodes, the triggers in both UGBA (Dai et al., 2023) and GTA (Xi et al., 2021) are still outliers. Thus, they propose a defense method called OD, which involves training a graph auto-encoder and filtering out nodes with high reconstruction loss. Their empirical results show that existing backdoor attack methods on graphs suffer from either a low ASR or outlier issues. However, there is no existing work on defending against DPGBA, which generates in-distribution triggers.

# B    ADDITIONAL EMPIRICAL RESULTS ON PREDICTION VARIANCE CAUSED BY EDGE DROPPING

In this section, we show additional results on prediction variance caused by dropping trigger edges and clean edges on Cora and PubMed datasets. We generate 40 triggers for Cora and 160 triggers for PubMed. The attack method used is DPGBA (Zhang et al., 2024). The model architecture is a 2-layer GCN (Kipf & Welling, 2016). The results are shown in Fig. 4. From the figure, it is evident that dropping adversarial edge connecting backdoor trigger will typically result in a much larger prediction variance than dropping clean edges.

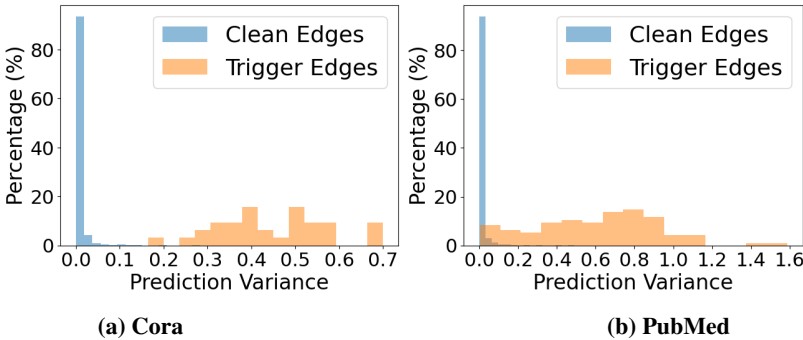

(a) Cora                              (b) PubMed

Figure 4: Visualization of prediction variance caused by dropping trigger edges and clean edges.

# C    TIME COMPLEXITY ANALYSIS

Following (Blakely et al., 2021), to show the computation of a GCN, first we define $\hat{\mathbf{A}} = \mathbf{A} + \mathbf{I}$, the adjacency matrix with self-loops. $\hat{\mathbf{D}}$ is the diagonal degree matrix of $\hat{\mathbf{A}}$. $\mathbf{A}'$ is the normalized adjacency matrix given as:

$$\mathbf{A}' = \hat{\mathbf{D}}^{-\frac{1}{2}} \hat{\mathbf{A}} \hat{\mathbf{D}}^{-\frac{1}{2}} \tag{7}$$

The computation of the $l$-th layer of a GCN network is:

$$\mathbf{X}^{(l+1)} = \sigma(\mathbf{A}' \mathbf{X}^{(l)} \mathbf{W}^{(l)}) \tag{8}$$

where $\sigma(\cdot)$ is a non-linear activation function (typically ReLU) and $\mathbf{W}^{(l)}$ is a feature transformation matrix $\in \mathbb{R}^{M_l \times M_{l+1}}$.

Thus, we analyze the complexity of the forward step by decomposing Equation 8 into three high-level operations:

1. $\mathbf{Z}^{(l)} = \mathbf{X}^{(l)} \mathbf{W}^{(l)}$: feature transformation
2. $\mathbf{X}^{(l+1)} = \mathbf{A}' \mathbf{Z}^{(l)}$: neighborhood aggregation
3. $\sigma(\cdot)$: activation

Part 1 is a dense matrix multiplication between matrices of size $N \times M_l$ and $M_l \times M_{l+1}$. We assume for all $l$, $M_l = M_{l+1} = M$. Therefore, this is $\mathcal{O}(NM^2)$.

Naively, part 2 involves multiplying matrices of sizes $N \times N$ and $N \times M$, resulting in a time complexity of $\mathcal{O}(N^2 M)$. However, in practice, we use a sparse operator like PyTorch's `scatter` function to compute this. For each row $(i, j)$ of the edge matrix $\mathbf{E}$, we compute $\mathbf{x}_i^{(l+1)} + = \mathbf{z}_j^{(l)}$, where $\mathbf{x}_i^{(l+1)}$ and $\mathbf{z}_j^{(l)}$ are $M$-dimensional vectors. This results in a total cost of $\mathcal{O}(|E|M)$, where $|E|$ is the total number of edges. Alternatively, considering each node has $d$ neighbors on average, neighborhood aggregation for each node requires $\mathcal{O}(dM)$, leading to a total cost of $\mathcal{O}(NdM) = \mathcal{O}(|E|M)$.

Part 3 is simply an element-wise function, so its cost is $\mathcal{O}(N)$.

Over $L$ layers, this results in time complexity $\mathcal{O}(L(NM^2 + NdM + N)) = \mathcal{O}(L(NM^2 + NdM)) = \mathcal{O}(L(NM^2 + |E|M))$. Next, we ignore the time complexity of activation for simplicity in our analysis.

**Random edge dropping:** For random edge dropping and conduct inference on the whole graph for $K$ times. The overall time complexity over $L$ layers is $\mathcal{O}(LK(NM^2 + |E|M))$.

**Dropping each edge individually:** For each node, we drop one of its neighbors and conduct inference on its $L$-hop subgraph centered at the node using an $L$-layer GNN, we repeat this process for $d$ times. For simplicity, we assume each node has $d$-neighbors. Thus, an $L$-hop subgraph has in total $\mathcal{O}(d^L)$ nodes. The time complexity analysis for each node is as follows. We use the $L$-hop subgraph as input to the $L$-layer GNN. In each GNN layer: (1) For feature transformation, the time complexity is $\mathcal{O}(d^L M^2)$. (2) For each neighbor aggregation, the time complexity is $\mathcal{O}(d^L M)$. As we conduct $d$ times aggregation, the time complexity is $\mathcal{O}(d^{L+1}M)$. Thus, over $N$ nodes and $L$-layer, the time complexity is $\mathcal{O}(LNd^L M(d + M))$.

As $d^L$ is usually much larger than $K$ in practice, the intuitive method is inefficient due to the high computational cost. We conduct empirical experiments to compare the running time on the OGB-arxiv dataset. We use the same experimental setup as described in Section 5.2. The results are shown in Tab. 4. We set $K = 10$ for RIGBD. From the table, we observe that RIGBD is significantly more efficient compared to the intuitive method in empirical experiments.

Table 4: Running Time Comparison

| **Intuitive** | **RIGBD** ($K = 10$) |
|---|---|
| 42.56s | 0.09s |

# D  DETAILED PROOFS

**Assumptions on Graphs.** Following (Dai et al., 2022b; Ma et al., 2021a), we consider a graph $\mathcal{G}$, where each node $v_i$ has features $\mathbf{x}_i \in \mathbb{R}^m$, a label $y_i$ and $\deg(i)$ denotes the number of its neighbors. We assume that: (1) The feature of node $v_i$ is sampled from the feature distribution $F_{y_i}$ that depends on the label $y_i$ of the node, i.e., $\mathbf{x}_i \sim \mathcal{F}_{y_i}$. This means that the feature of the node is influenced by its label, with $\mu(\mathcal{F}_{y_i})$ denoting the mean of the distribution. (2) Feature dimensions of $\mathbf{x}_i$ are independent to each other; (3) The features in $\mathbf{X}$ are bounded by a positive scalar $B$, i.e., $\max_{i,j} |\mathbf{X}[i,j]| \leq B$;

These assumptions are reasonable in the context of graph machine learning for several reasons: Feature Distribution Based on Labels (Assumption 1): In many real-world scenarios, the features of a node are influenced by its label. For example, in a social network, the attributes of a user (like interests or activities) are often correlated with their group or community (label). Independence of Feature Dimensions (Assumption 2): In many datasets, especially high-dimensional ones, the correlations between features may be weak or negligible. Bounded Features (Assumption 3): In real-world datasets, feature values are often bounded due to physical, biological, or other practical constraints. Furthermore, in practice, features are often normalized or standardized during preprocessing, effectively bounding them within a certain range.

## D.1  PROOF OF THEOREM 1

**Theorem 1.** *Consider a graph $\mathcal{G} = \{\mathcal{V}, \mathcal{E}, \mathbf{X}\}$ following Assumptions (1)-(3). For clean node $v_i \in \mathcal{V}$ and its neighbors $\mathcal{N}(i)$, the expectation of the pre-activation output of a single operation defined in Eq. 3 is given by $\mathbb{E}[\mathbf{h}_i]$. Then the expectation of the pre-activation output of a single operation defined in Eq. 3 after random edge dropping is given by $\mathbb{E}[\mathbf{h}_i^k] = \mathbb{E}[\mathbf{h}_i]$.*

*Proof.* Given the graph convolution operation defined in Eq. 3, we obtain node representation $\mathbb{E}[\mathbf{h}_i] = \mathbb{E}\left[\sum_{j \in \mathcal{N}(i)} \frac{1}{\sqrt{deg(i)}\sqrt{deg(j)}} \mathbf{W}\mathbf{x}_j\right]$ For any node $v_i \in \mathcal{V}$ and its neighbors $\mathcal{N}(i)$, after random edge dropping with drop ratio $\beta$, in expectation, each neighbor in $\mathcal{N}(i)$ contributes with a weight of $(1 - \beta)$, and the expected $\deg(i)$ is also scaled by $(1 - \beta)$. Therefore, the expectation of the pre-activation output of a single operation defined in Eq. 3 after random edge dropping is given

by

$$
\begin{aligned}
\mathbb{E}[\mathbf{h}_i^k] &= \mathbb{E}\left[\sum_{j \in \mathcal{N}(i)} \frac{1}{\sqrt{(1-\beta)deg(i)}\sqrt{(1-\beta)deg(j)}}\mathbf{W}(1-\beta)\mathbf{x}_j\right] \\
&= \mathbb{E}\left[\sum_{j \in \mathcal{N}(i)} \frac{1}{(1-\beta)\sqrt{(deg(i)}\sqrt{deg(j)}}\mathbf{W}(1-\beta)\mathbf{x}_j\right] = \mathbb{E}[\mathbf{h}_i],
\end{aligned}
\tag{9}
$$

which completes the proof. □

## D.2    PROOF OF THEOREM 2

To prove Theorem 2, we first introduce the celebrated Hoeffding inequality.

**Lemma 1. (Hoeffding's Inequality).** *Let $Z_1, \ldots, Z_n$ be independent bounded random variables with $Z_i \in [a, b]$ for all $i$, where $-\infty < a \le b < \infty$. Then*

$$
\mathbb{P}\left(\frac{1}{n}\sum_{i=1}^n (Z_i - \mathbb{E}[Z_i]) \ge t\right) \le \exp\left(-\frac{2nt^2}{(b-a)^2}\right)
$$

*and*

$$
\mathbb{P}\left(\frac{1}{n}\sum_{i=1}^n (Z_i - \mathbb{E}[Z_i]) \le -t\right) \le \exp\left(-\frac{2nt^2}{(b-a)^2}\right)
$$

*for all $t \ge 0$.*

**Theorem 2.** *Consider a graph $\mathcal{G} = \{\mathcal{V}, \mathcal{E}, \mathbf{X}\}$ following Assumptions (1)-(3). Let $\mathbf{h}_i$ and $\mathbf{h}_i^k$ denote the clean node embedding before and after the $k$-th random edge dropping, respectively, and $\deg(i)_k$ represent the corresponding number of remaining neighbors. The probability that the distance between the observation $\mathbf{h}_i^k$ and the expectation of $\mathbf{h}_i$ is larger than $t$ is bounded by:*

$$
\mathbb{P}\left(\left\|\mathbf{h}_i^k - \mathbb{E}[\mathbf{h}_i]\right\|_2 \ge t\right) \le 2 \cdot M \cdot \exp\left(-\frac{\deg(i)_k t^2}{2\rho^2(\mathbf{W})B^2 M}\right),
\tag{10}
$$

*where $M$ denotes the feature dimensionality and $\rho^2(\mathbf{W})$ denotes the largest singular value of $\mathbf{W}$.*

*Proof.* Following (Dai et al., 2022b; Ma et al., 2021a), we consider a $d$-regular graph $\mathcal{G}$, i.e. each node has $d$ neighbors, then the expectation of $\mathbf{h}_i$ can be derived as $\mathbb{E}[\mathbf{h}_i] = \mathbb{E}\left[\sum_{j \in \mathcal{N}(i)} \frac{1}{deg(i)}\mathbf{W}\mathbf{x}_j\right]$. Let $\mathbf{x}_j^k[m], n = 1, \ldots, M$ denote the $m$-th element of $\mathbf{x}_j^k$. Then, for any dimension $m$, $\{x_j^k[m], j \in \mathcal{N}(i)\}$ is a set of independent random variables bounded by $[-B, B]$. Hence, directly applying Hoeffding's inequality, for any $t_1 \ge 0$, we have the following bound:

$$
\mathbb{P}\left(\left|\frac{1}{\mathcal{N}(i)^k}\sum_{j \in \mathcal{N}(i)^k}\left(\mathbf{x}_j^k[m] - \mathbb{E}\left[\mathbf{x}_j^k[m]\right]\right)\right| \ge t_1\right) \le 2\exp\left(-\frac{(\deg(i))_k t_1^2}{2B^2}\right)
\tag{11}
$$

if $\left\|\frac{1}{\mathcal{N}(i)^k}\sum_{j \in \mathcal{N}(i)^k}\left(\mathbf{x}_j^k - \mathbb{E}\left[\mathbf{x}_j^k\right]\right)\right\|_2 \ge \sqrt{M}t_1$, then at least for $m \in \{1, ..., M\}$, the inequality $\left|\frac{1}{\mathcal{N}(i)^k}\sum_{j \in \mathcal{N}(i)^k}\left(\mathbf{x}_j^k[m] - \mathbb{E}\left[\mathbf{x}_j^k[m]\right]\right)\right| \ge t_1$ holds. Hence, we have

$$
\begin{aligned}
\mathbb{P}\left(\left\|\frac{1}{\mathcal{N}(i)^k}\sum_{j \in \mathcal{N}(i)^k}\left(\mathbf{x}_j - \mathbb{E}\left[\mathbf{x}_j\right]\right)\right\|_2 \ge \sqrt{M}t_1\right) &\le \mathbb{P}\left(\bigcup_{m-1}^M\left\{\left|\frac{1}{\mathcal{N}(i)^k}\sum_{j \in \mathcal{N}(i)^k}\left(\mathbf{x}_j^k[m] - \mathbb{E}\left[\mathbf{x}_j^k[m]\right]\right)\right| \ge t_1\right\}\right) \\
&\le \sum_{m=1}^M \mathbb{P}\left(\left|\frac{1}{\mathcal{N}(i)^k}\sum_{j \in \mathcal{N}(i)^k}\left(\mathbf{x}_j^k[m] - \mathbb{E}\left[\mathbf{x}_j^k[m]\right]\right)\right| \ge t_1\right) \\
&= 2 \cdot l \cdot \exp\left(-\frac{(\deg(i)_k)t_1^2}{2B^2}\right)
\end{aligned}
\tag{12}
$$

Let $t1 = \frac{t2}{\sqrt{M}}$, then we have

$$\mathbb{P}\left(\left\|\frac{1}{\mathcal{N}(i)^k}\sum_{j\in\mathcal{N}(i)^k}\left(\mathbf{x}_j - \mathbb{E}\left[\mathbf{x}_j\right]\right)\right\|_2 \geq t_2\right) \leq 2\cdot M\cdot\exp\left(-\frac{(\deg(i)_k)t_1^2}{2B^2M}\right) \tag{13}$$

Furthermore, we have

$$
\begin{aligned}
\left\|\mathbf{h}_i^k - \mathbb{E}\left[\mathbf{h}_i^k\right]\right\|_2 &= \left\|\mathbf{W}\left(\frac{1}{\mathcal{N}(i)^k}\sum_{j\in\mathcal{N}(i)^k}\left(\mathbf{x}_j^k - \mathbb{E}\left[\mathbf{x}_j^k\right]\right)\right)\right\|_2 \\
&\leq \left\|\mathbf{W}\right\|_2\left\|\frac{1}{\mathcal{N}(i)^k}\sum_{j\in\mathcal{N}(i)^k}\left(\mathbf{x}_j^k - \mathbb{E}\left[\mathbf{x}_j^k\right]\right)\right\|_2 \\
&= \rho(\mathbf{W})\left\|\frac{1}{\mathcal{N}(i)^k}\sum_{j\in\mathcal{N}(i)^k}\left(\mathbf{x}_j^k - \mathbb{E}\left[\mathbf{x}_j^k\right]\right)\right\|_2,
\end{aligned}
\tag{14}
$$

where $\left\|\mathbf{W}\right\|_2$ is the matrix 2-norm of $\mathbf{W}$. We denote $\rho(\mathbf{W}) = \left\|\mathbf{W}\right\|_2$. Given $\mathbb{E}\left[\mathbf{h}_i\right] = \mathbb{E}\left[\mathbf{h}_i^k\right]$, we have

$$
\begin{aligned}
\mathbb{P}\left(\left\|\mathbf{h}_i^k - \mathbb{E}\left[\mathbf{h}_i\right]\right\|_2 \geq t\right) &= \mathbb{P}\left(\left\|\mathbf{h}_i^k - \mathbb{E}\left[\mathbf{h}_i^k\right]\right\|_2 \geq t\right) \\
&\leq \mathbb{P}\left(\rho(\mathbf{W})\left\|\frac{1}{\mathcal{N}(i)^k}\sum_{j\in\mathcal{N}(i)^k}\left(\mathbf{x}_j^k - \mathbb{E}\left[\mathbf{x}_j^k\right]\right)\right\|_2 \geq t\right) \\
&= \mathbb{P}\left(\left\|\frac{1}{\mathcal{N}(i)^k}\sum_{j\in\mathcal{N}(i)^k}\left(\mathbf{x}_j^k - \mathbb{E}\left[\mathbf{x}_j^k\right]\right)\right\|_2 \geq \frac{t}{\rho(\mathbf{W})}\right) \\
&\leq 2\cdot M\cdot\exp\left(-\frac{\deg(i)_k t^2}{2\rho^2(\mathbf{W})B^2M}\right),
\end{aligned}
\tag{15}
$$

which completes the proof. $\qquad\square$

### D.3 PROOF OF THEOREM 3

**Theorem 3.** *Without loss of generality, we consider the case where there is only one edge connecting a backdoor trigger to a target node. Let $\beta$ denote the random edge dropping ratio, and $K$ be the total number of iterations for random edge dropping and conducting inference on the perturbed graph. For a poisoned node $v_i$ and a trigger $g_i$, the expected number of times the poisoned node $v_i$ demonstrates large prediction variance is given as $\mathbb{E}(g_i)_d = K\cdot\beta$.*

*Proof.* We model the random edge-dropping process as a series of independent Bernoulli trials. Each trial represents a single iteration of edge-dropping, with a success probability $\beta$, which is the probability that the edge connecting $v_i$ and $g_i$ is dropped in that iteration.

Let $X_k$ be an indicator random variable for the $k$-th iteration, where $X_k = 1$ if the edge is dropped, and $X_k = 0$ otherwise. The total number of times the edge is dropped over $K$ iterations is then the sum of these indicator variables:

$$X = \sum_{k=1}^{K} X_k \tag{16}$$

Since each $X_k$ follows a Bernoulli distribution with parameter $\beta$, the sum $X$ follows a Binomial distribution with parameters $K$ and $\beta$:

$$X \sim \text{Binomial}(K, \beta) \tag{17}$$

The expected value of a Binomial distribution is given by the product of the number of trials and the success probability. Therefore, the expected number of times the poisoned node $v_i$ shows large prediction variance is:

$$E(g_i)_d = K \times \beta, \tag{18}$$

which completes the proof. $\qquad\square$

# E   EMPIRICAL RESULTS OF PREDICTION VARIANCE FOR DIFFERENT ATTACKS AND DATASETS USING RANDOM EDGE DROPPING

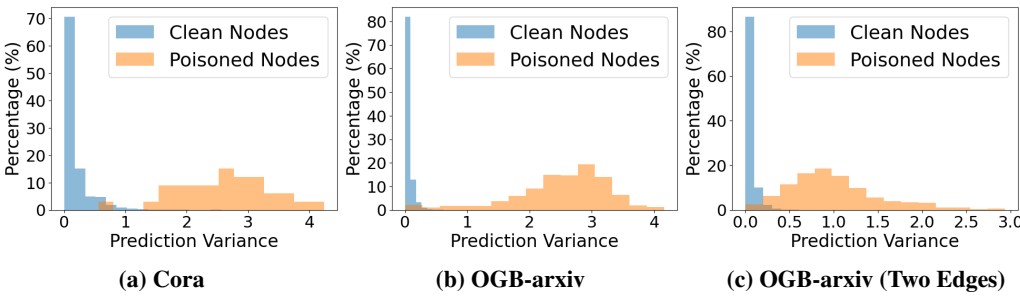

| (a) Cora | (b) OGB-arxiv | (c) OGB-arxiv (Two Edges) |

Figure 5: Visualization of prediction variance caused by random edge dropping.

To empirically verify that our method of random edge dropping can efficiently distinguish poisoned nodes from clean nodes, we conduct experiments on the Cora (Sen et al., 2008), PubMed, and OGB-arxiv (Hu et al., 2020) datasets, using 40 triggers for Cora, 160 triggers for PubMed and 565 triggers for OGB-arxiv, respectively. The attack method used is DPGBA (Zhang et al., 2024). The model architecture is a 2-layer GCN (Kipf & Welling, 2016). We set the drop ratio $\beta = 0.5$ and performed $K = 10$ iterations of random edge dropping. The results are shown in Fig. 5. In Figs. 5(a) and (b), we show the results for Cora and OGB-arxiv, respectively. In Fig. 5(c), we show the result for OGB-arxiv with two edges linking a backdoor trigger and a target node. Our observations are as follows: **(i)** Our method consistently results in higher prediction variance for poisoned nodes, thus enabling the distinction between poisoned nodes and clean nodes. **(ii)** Even when two edges link a backdoor trigger to a poisoned node, our method still results in high prediction variance for most of the poisoned nodes. This demonstrates the superior performance of our method compared to the intuitive approach of dropping each edge individually. Additional visualizations of prediction variance under different attacks on various datasets are in Fig. 6, Fig. 7 and Fig 8.

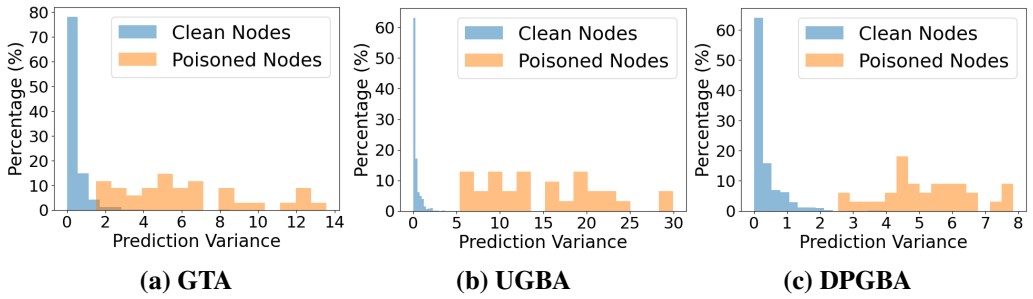

| (a) GTA | (b) UGBA | (c) DPGBA |

Figure 6: Visualization of prediction variance caused by random edge dropping on Cora dataset, drop ratio $\beta = 0.5$, drop iterations $K = 20$ and the number of triggers is 40.

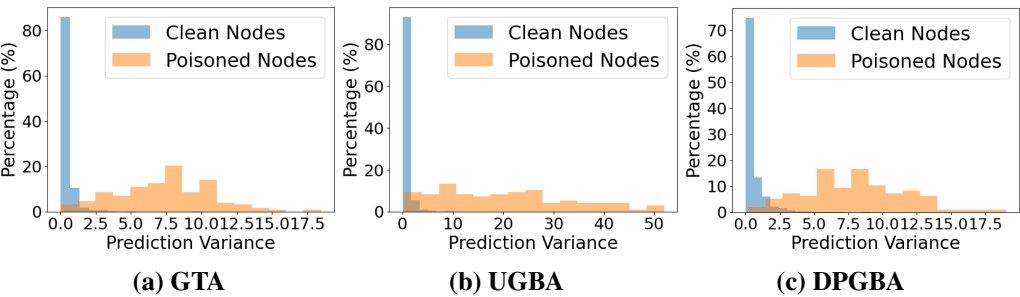

| (a) GTA | (b) UGBA | (c) DPGBA |

Figure 7: Visualization of prediction variance caused by random edge dropping on PubMed dataset, drop ratio $\beta = 0.5$, drop iterations $K = 20$ and the number of triggers is 160.

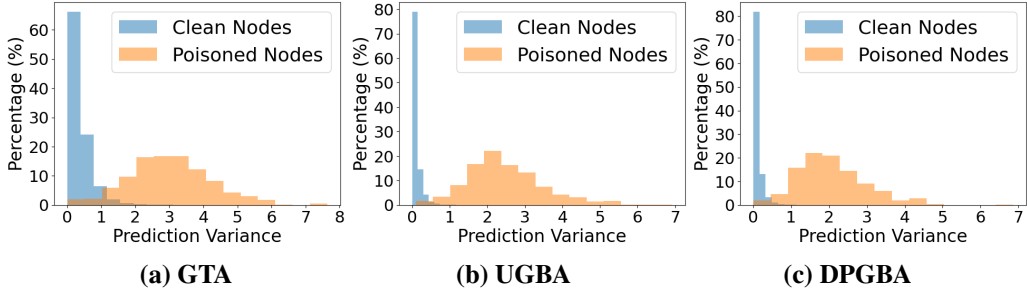

Figure 8: Visualization of prediction variance caused by random edge dropping on OGB-arxiv dataset, drop ratio $\beta = 0.5$, drop iterations $K = 20$ and the number of triggers is 565.

## F EMPIRICAL RESULTS OF PREDICTION VARIANCE FOR NODES WITH LOW DEGREES AND POISONED NODES

In this section, to address concerns about high prediction variance in low-degree or heterophily nodes after random edge dropping, we compare the prediction variance of nodes from both homophily and heterophily graphs against poisoned nodes. We utilize the UGBA attack method to generate backdoor triggers for each dataset, setting $K$ to 10 and the drop ratio to 0.5. For each dataset, we select nodes with degrees 1, 2, and 3. Our experiments are conducted on homophily graphs (Cora, Citeseer, Pubmed, Physics, OGB-arxiv) and heterophily graphs (Wisconsin, Cornell, Texas (Pei et al., 2020), Flickr). The results, shown in Table 5, reveal that for heterophily graphs such as Wisconsin, Cornell, and Texas, the prediction variance gap between low-degree and poisoned nodes is smaller compared to homophily graphs. However, the prediction variance of poisoned nodes remains significantly higher than that of clean nodes, even for those with low degrees. This suggests that a backdoored GNN model shows high confidence in the target class when a node is linked to a backdoor trigger, but when the trigger is removed, it shifts to high confidence in another class. In contrast, for clean nodes, edge dropping typically does not cause a significant shift in prediction confidence.

Table 5: Comparison of prediction variance between clean nodes with low degrees and poisoned nodes.

| Dataset | Degree-1 | Degree-2 | Degree-3 | Target Nodes |
|---------|----------|----------|----------|--------------|
| Wisconsin | 0.64 | 1.94 | 1.48 | 6.08 |
| Cornell | 0.79 | 1.33 | 1.27 | 5.16 |
| Texas | 0.65 | 1.48 | 1.28 | 4.10 |
| Cora | 0.51 | 0.66 | 0.56 | 12.70 |
| Citeseer | 0.36 | 0.70 | 0.69 | 5.41 |
| Pubmed | 0.67 | 0.80 | 0.57 | 10.67 |
| Physics | 0.44 | 0.52 | 0.36 | 18.80 |
| Flickr | 0.01 | 0.02 | 0.01 | 2.56 |
| OGB-arxiv | 0.57 | 0.79 | 0.67 | 10.67 |

## G TRAINING ALGORITHM

We summarize the training method of RIGBD for training a backdoor robust GNN node classifier in Algorithm 1. Specifically, we start by randomly initializing the parameters $\theta_b$ for an L-layer GNN node classifier $f_b$, which uses the graph convolution operation defined in Eq. 3 (line 1). In each iteration of the training loop (lines 2-4), we update $\theta_b$ by training $f_b$ on the backdoored graph $\mathcal{G}_T$ using supervised learning until the model converges. After convergence, we enter a second loop (lines 5-8) where we conduct random edge dropping on $\mathcal{G}_T$ and perform inference with the backdoor model $f_b$ (line 6). For each node, we calculate the prediction variance $s(i)$ using Eq. 2 (line 7). Nodes with prediction variance $s(i) > \tau$ are selected as candidate poisoned nodes $\mathcal{V}_s$, where $\tau$ is calculated by Eq. 5. Next, we randomly initialize $\theta$ for another L-layer GNN node classifier $f$ (line 10). In the final training loop (lines 11-13), we update $\theta$ by training $f$ using the robust training strategy defined in Eq.

6 until convergence. The algorithm concludes by returning the backdoor robust GNN node classifier $f$ (line 14).

---

**Algorithm 1** Algorithm of RIGBD

---

**Require:** Backdoored graph $\mathcal{G}_T = (\mathcal{V}_T, \mathcal{E}_T, \mathbf{X}_T)$, drop ratio $\beta$ and number of drop iterations $K$
**Ensure:** Backdoor robust node classifier $f$
 1: Randomly initialize $\theta_b$ for a $L$-layer GNN node classifier $f_b$ with graph convolution operation defined in Eq. 3;
 2: **while** not converged yet **do**
 3:     Update $\theta_b$ by training $f_b$ on the backdoored graph $\mathcal{G}_T$ using supervised learning;
 4: **end while**
 5: **for** $k = 1, 2, \ldots, K$ **do**
 6:     Conduct random edge dropping on the graph $\mathcal{G}_T$ and perform inference with the backdoor model $f_b$;
 7:     Calculate the prediction variance $s(i)$ for each node by Eq. 2;
 8: **end for**
 9: Select candidate poisoned nodes $\mathcal{V}_s$ for nodes with prediction variance $s(i) > \tau$, where $\tau$ is calculated by Eq. 5.
10: Randomly initialize $\theta$ for any $L$-layer GNN node classifier $f$;
11: **while** not converged yet **do**
12:     Update $\theta$ by training $f$ using the robust training strategy defined in Eq. 6;
13: **end while**
14: **return** Backdoor robust GNN node classifier $f$;

---

## H  ADDITIONAL DETAILS OF EXPERIMENT SETTINGS

### H.1  DATASET STATISTICS

**Cora, Citeseer and PubMed** are citation networks where nodes denote papers, and edges depict citation relationships. In Cora and Citeseer, each node is described using a binary word vector, indicating the presence or absence of a corresponding word from a predefined dictionary. In contrast, PubMed employs a TF/IDF weighted word vector for each node. For both datasets, nodes are categorized based on their respective research areas.

**Coauther Physics** is a co-authorship graph based on the Microsoft Academic Graph from the KDD Cup 2016 challenge (Sinha et al., 2015). Nodes are authors, that are connected by an edge if they co-authored a paper; node features represent paper keywords for each author's papers, and class labels indicate most active fields of study for each author.

**Flicker** (Zeng et al., 2019): In this graph, each node symbolizes an individual image uploaded to Flickr. An edge is established between the nodes of two images if they share certain attributes, such as geographic location, gallery, or user comments. The node features are represented by a 500-dimensional bag-of-word model provided by NUS-wide. Regarding labels, we examined the 81 tags assigned to each image and manually consolidated them into 7 distinct classes, with each image falling into one of these categories.

**OGB-arxiv** is a citation network encompassing all Computer Science arXiv papers cataloged in the Microsoft Academic Graph. Each node is characterized by a 128-dimensional feature vector, which is derived by averaging the skip-gram word embeddings present in its title and abstract. Additionally, the nodes are categorized based on their respective research areas. The statistics details of these datasets are summarized in Table 6.

Table 6: Dataset Statistics

| Datasets | #Nodes | #Edges | #Features | #Classes |
|----------|--------|--------|-----------|----------|
| Cora | 2,708 | 5,429 | 1,443 | 7 |
| Citeseer | 3,327 | 4,552 | 3,703 | 3 |
| Pubmed | 19,717 | 44,338 | 500 | 3 |
| Physics | 34,493 | 495,924 | 8,415 | 5 |
| Flickr | 89,250 | 899,756 | 500 | 7 |
| OGB-arxiv | 169,343 | 1,166,243 | 128 | 40 |

## H.2 ATTACK METHODS

The details of attack methods are described following:

1. **GTA** (Xi et al., 2021): GTA utilizes a trigger generator that crafts subgraphs as triggers tailored to individual samples. The optimization of the trigger generator focuses exclusively on the backdoor attack loss, disregarding any constraints related to trigger detectability.

2. **UGBA** (Dai et al., 2023): UGBA selects representative and diverse nodes as poisoned nodes to fully utilize the attack budget. An adaptive trigger generator is optimized with a constraint loss so that the generated triggers are ensured to be similar to the target nodes.

3. **DPGBA** (Dai et al., 2023): DPGBA introduce adversarial learning strategy to generate in-distribution triggers. A novel loss is proposed to help adaptive trigger generator to generate efficient in-distribution triggers.

## H.3 COMPARED METHODS

We select two defense methods that aim to defend against specific backdoor attack methods:

1. **Prune** (Dai et al., 2023): Prune is designed to prune edges that linking nodes with low similarity to remove those triggers that are totally different from clean nodes.

2. **OD** (Zhang et al., 2024): OD introduces graph auto-encoder and filter out nodes with high reconstruction loss to remove those triggers that are outliers compared to clean ones.

We include a strong baseline that also aims to learn a clean model from the poisoned data:

1. **ABL** (Li et al., 2021a): ABL exploits two key weaknesses of backdoor attacks: 1) models learn backdoored data much faster than clean data, with stronger attacks causing faster convergence; 2) the backdoor task is tied to a specific class. Thus, they propose a local gradient ascent (LGA) technique to trap the loss value of each example around a certain threshold. They also propose a global gradient ascent (GGA) loss function to unlearn the backdoor with a small subset of backdoor examples while continuing to learn from the remaining clean examples.

As backdoor attack is a subset of a poisoning attack, we also include three representative robust GNNs:

1. **RS** (Wang et al., 2021): Randomized smoothing on graphs was first proposed to counter adversarial structural perturbations. We adopt this method and set the drop ratio $\beta > 0.5$ to balance the defense performance and clean accuracy. To further ensure clean accuracy, we introduce an adversarial training strategy during the training phase.

2. **GNNGuard** (Zhang & Zitnik, 2020): GNNGuard is a robust defense method for GNNs that protects against adversarial attacks by leveraging node similarity to filter out adversarial edges. It employs a multi-stages defense strategy that dynamically adjusts edge weights during training, enhancing the model's resilience to structural perturbations.

3. **RobustGCN** (Zhu et al., 2019): RobustGCN enhances the robustness of GCN against adversarial attacks by using Gaussian distributions as node representations, which absorb the effects of adversarial changes. It also introduces a variance-based attention mechanism

that assigns different weights to node neighborhoods based on their variances, reducing the propagation of adversarial effects through the graph.

## H.4 DETAILED COMPARISON OF RIGBD TO BASELINE DEFENSE METHODS

Aside from the defense methods we mentioned, such as Prune and OD, which aim to defend against attack methods by exploiting the characteristics of generated triggers, to the best of our knowledge, our RIGBD is the first work that specifically aims to defend against dirty-label node-level graph backdoor attacks that involve generating a backdoor trigger and linking it to the target node from the perspective of training a backdoor-robust model.

Since such backdoor attack methods inherently belong to the category of poisoning attacks, we include GNNGuard and RobustGCN as baselines because they both have the ability to mitigate the influence of malicious neighbors, which may reduce the impact of backdoor triggers as these triggers are neighbors of the target nodes. However, their defense performance is not satisfactory, especially when the generated triggers are similar to the target nodes.

We also include randomized smoothing because if we can randomly drop edges with a high drop ratio during multiple inferences, we could obtain the final prediction as if the backdoor trigger were dropped. However, the clean accuracy of this method cannot be guaranteed, as too many edges are dropped during the classification task.

The core idea of ABL is that the model quickly learns backdoored data, after which a local gradient ascent technique is used to trap the loss value of each example. However, their framework requires manual tuning of the isolation ratio, and a high isolation ratio inevitably leads to a performance drop on clean data when their losses are trapped. This significantly reduces the method's utility in real-world applications, as defenders typically do not know the number of poisoned target nodes.

In contrast, (i) Our method is theoretically guaranteed and capable of identifying those poisoned target nodes with high precision without knowledge of the number of poisoned target nodes. (ii) With high precision in detecting poisoned nodes, our framework makes only minimal changes to the graph structure of clean nodes, ensuring that the clean accuracy remains comparable to that of a model trained on a clean graph. (iii) With the proposed robust training strategy, our framework is robust to different hyperparameter selections, making it easy to implement and enhancing its utility in real-world applications.

## H.5 IMPLEMENTATION DETAILS

We test our RIGBD using different graph neural network backbones, i.e. GCN, GAT, and GraphSage. All hyperparameters of all methods are tuned based on the validation set for fair comparison. All models are trained on an A6000 GPU with 48G memory.

## I RESULTS ON CLEAN GRAPH

Table 7: Comparison of Clean Accuracy between GCN and RIGBD trained on clean graph.

| Dataset | GCN | RIGBD |
|---|---|---|
| Cora | 84.44 | 84.07 |
| Citeseer | 73.83 | 73.44 |
| PubMed | 85.13 | 85.05 |
| Physics | 95.78 | 95.40 |
| Flickr | 44.42 | 44.03 |
| OGB-arxiv | 66.13 | 65.53 |

In this section, we conduct experiments to show the clean accuracy of our RIGBD when trained on a clean graph. Specifically, we follow the experimental setup described in Sec. 5.2 and remove the backdoor triggers from the poisoned graph. Then, we train a GCN and RIGBD on this clean graph. The results are shown in Tab. 7. From the table, we observe that our RIGBD achieves comparable

Table 8: Additional results of backdoor defense.

| Attacks | Defense | Cora | | PubMed | | OGB-arxiv | |
|---|---|---|---|---|---|---|---|
| | | ASR(%)↓ | ACC(%)↑ | ASR(%)↓ | ACC(%)↑ | ASR(%)↓ | ACC(%)↑ |
| GTA | GAT | 76.97 | 84.44 | 89.77 | 82.60 | OOM | OOM |
| | GraphSage | 100.00 | 81.11 | 99.94 | 84.88 | 93.67 | 67.81 |
| | RS-0.7 | 45.02 | 66.67 | 33.20 | 85.24 | 33.12 | 56.93 |
| | RS-0.8 | 34.32 | 65.56 | 24.65 | 85.13 | 25.72 | 55.05 |
| | RIGBD-GAT | 0.02 | 85.19 | 0.08 | 83.66 | OOM | OOM |
| | RIGBD-GraphSage | 0.18 | 81.52 | 0.00 | 85.84 | 0.09 | 67.18 |
| UGBA | GAT | 100 | 85.19 | 100.00 | 83.05 | OOM | OOM |
| | GraphSage | 97.33 | 84.07 | 99.08 | 85.24 | 94.69 | 68.25 |
| | RS-0.7 | 46.86 | 69.63 | 37.74 | 85.34 | 29.74 | 57.33 |
| | RS-0.8 | 37.64 | 65.19 | 28.20 | 86.35 | 23.14 | 54.77 |
| | RIGBD-GAT | 0.04 | 84.81 | 0.30 | 82.14 | OOM | OOM |
| | RIGBD-GraphSage | 0.44 | 84.07 | 0.02 | 85.74 | 0.06 | 67.75 |
| DPGBA | GAT | 91.14 | 83.33 | 93.8 | 83.56 | OOM | OOM |
| | GraphSage | 93.48 | 83.33 | 91.8 | 85.79 | 99.53 | 67.76 |
| | RS-0.7 | 40.96 | 68.52 | 36.31 | 84.98 | 32.81 | 56.66 |
| | RS-0.8 | 35.79 | 66.30 | 27.13 | 85.74 | 23.41 | 54.45 |
| | RIGBD-GAT | 0.04 | 84.44 | 0.01 | 83.31 | OOM | OOM |
| | RIGBD-GraphSage | 0.26 | 82.11 | 0.06 | 86.35 | 0.03 | 66.61 |

clean accuracy to GCN. This is because our poisoned node selection strategy, defined in Eq. 5, stops when nodes continuously show different labels compared to the node with the highest prediction variance. Since nodes with high confidence are not always from the same class in a clean graph, we rarely select nodes as poisoned nodes. This highlights the adaptability of our RIGBD, even when we do not know whether the graph is clean or poisoned.

## J   ADDITIONAL RESULTS OF DEFENSE PERFORMANCE

In this section, we provide additional results on defense performance. Specifically, we implement RIGBD with GAT and GraphSage as backbones. We also include randomized smoothing (RS) with drop ratios $\beta = 0.7$ and $\beta = 0.8$. All other settings are the same as Sec. 5.2. The results are shown in Table 8. From the table, we observe the following: **(i)** With the increase in drop ratio, randomized smoothing can degrade the ASR. However, the clean accuracy is not guaranteed. This demonstrates the superiority of our framework in degrading ASR while maintaining clean accuracy. **(ii)** Our RIGBD consistently achieves low ASR while maintaining comparable or slightly better clean accuracy compared to vanilla GNNs on all datasets. This demonstrates the flexibility and effectiveness of our framework with different backbones.

## K   ADDITIONAL RESULTS OF DEFENSE PERFORMANCE ON HETEROPHILIC GRAPHS

| Dataset | Methods | Clean ACC | ASR | Recall | Precision |
|---|---|---|---|---|---|
| Wisconsin | GCN | 48 | 100 | - | - |
| | RIGBD | 48 | 0 | 90 | 95 |
| Cornell | GCN | 50.0 | 93.3 | - | - |
| | RIGBD | 53.5 | 3.3 | 70 | 100 |
| Texas | GCN | 44.4 | 92.9 | - | - |
| | RIGBD | 42.9 | 3.7 | 80 | 70 |

Table 9: Performance comparison of GCN and RIGBD across different datasets.

In this section, we conduct additional experiments to evaluate the performance of RIGBD on heterophilic graphs. The attack method used is DPGBA Zhang et al. (2024), with 20 poisoned nodes during training, and other settings follow those described in Section 5.1. The results are presented in Table 9. It is evident that RIGBD achieves consistently strong performance in defending against

Table 10: Results for the ability to detect poisoned nodes.

| Datasets | Attacks | Clean ACC | ASR | ACC | Recall | Precision |
|---|---|---|---|---|---|---|
| | GTA | 83.5 | 0.00 | 83.70 | 91.2 | 94.0 |
| Cora | UGBA | 83.5 | 0.01 | 84.81 | 96.8 | 100.0 |
| | DPGBA | 83.5 | 0.01 | 85.19 | 100.0 | 100.0 |
| | GTA | 73.8 | 0.34 | 74.10 | 94.7 | 100.0 |
| Citeseer | UGBA | 73.8 | 0.00 | 73.80 | 93.4 | 98.6 |
| | DPGBA | 73.8 | 0.33 | 73.79 | 97.4 | 100.0 |
| | GTA | 84.9 | 0.01 | 84.32 | 82.0 | 90.5 |
| PubMed | UGBA | 84.9 | 0.01 | 85.13 | 88.7 | 96.6 |
| | DPGBA | 84.9 | 0.01 | 84.32 | 85.0 | 91.0 |
| | GTA | 95.8 | 0.32 | 95.10 | 86.2 | 91.8 |
| Physics | UGBA | 95.8 | 0.12 | 95.71 | 96.0 | 98.4 |
| | DPGBA | 95.8 | 0.21 | 95.97 | 96.8 | 99.2 |
| | GTA | 44.4 | 0.00 | 44.21 | 69.5 | 98.1 |
| Flickr | UGBA | 44.4 | 0.00 | 42.74 | 98.6 | 97.2 |
| | DPGBA | 44.4 | 0.00 | 43.80 | 97.9 | 100.0 |
| | GTA | 65.8 | 0.01 | 66.51 | 84.9 | 90.3 |
| OGB-arxiv | UGBA | 65.8 | 0.01 | 65.21 | 95.6 | 96.3 |
| | DPGBA | 65.8 | 0.00 | 65.24 | 90.5 | 93.9 |

backdoor attacks while maintaining clean accuracy. The recall and precision of poisoned node detection further demonstrate that our method remains robust when applied to heterophilic graphs.

## L ADDITIONAL RESULTS OF THE ABILITY TO DETECT POISONED NODES

In this section, we provide additional results of the ability to detect poisoned nodes on Cora and PubMed dataset. We present the recall and precision of RIGBD in identifying poisoned nodes. We use the same setting as in Sec. 5.2. The results are shown in Table 10. We also present the corresponding ASR and ACC to illustrate the correlation between detection ability and defense performance. Clean accuracy (Clean ACC), obtained from a model trained on the clean graph, is provided as a reference to evaluate the model's performance on clean nodes. From the table, we observe the following: **(i)** Our RIGBD demonstrates consistently high precision, always over $90\%$, in detecting poisoned nodes across three attack methods in all datasets, with detection recall always exceeding $80\%$. This indicates that our strategy of random edge dropping with graph convolution operation defined in Eq. 3 typically leads to higher prediction variance for poisoned nodes. **(ii)** Although the recall for some attack methods and datasets is less than $90\%$, we still achieve an ASR close to $0\%$ while maintaining ACC. This demonstrates the stability of our strategy to train a robust node classifier using Eq. 6. Even when some of the poisoned nodes are not identified, it still efficiently helps the model unlearn the triggers.

## M ADDITIONAL RESULTS OF HYPERPARAMETER ANALYSIS

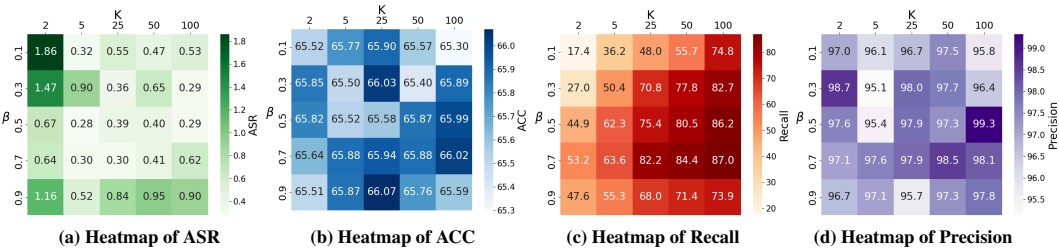

Figure 9: Hyperparameter Sensitivity Analysis

In this section, we provide additional results on how different drop ratios ($\beta$) and different numbers of iterations of random edge dropping ($K$) impact the ACC of RIGBD. ASR, Recall, and Precision are also provided as references. Specifically, we vary the values of $K$ as $\{2, 5, 25, 50, 100\}$, and the values of $\beta$ as $\{0.1, 0.3, 0.5, 0.7, 0.9\}$. The attack method used is DPGBA. The other settings are the same as the evaluation protocol in Sec. 5.1. The results are shown in Fig. 9. From the figure, we make the following observations: **(i)** Our RIGBD is stable in terms of ASR, ACC, and Precision. Notably, when the drop ratio $\beta = 0.1$ and we only conduct two iterations of random edge dropping, the recall

of our poison node detection is around 17.4%. However, we still achieve robust defense performance with ASR close to 0%. This demonstrates that our robust training strategy is effective even with a lower recall. As our framework identifies the most efficient triggers that lead to large prediction variance and focuses on minimizing their impact. By doing so, the influence of less efficient triggers is inherently counteracted. In contrast, when we simply remove those 17.4% triggers from the dataset and train a model on the remaining dataset, the ASR is around 80%. **(ii)** As $K$ increases, the recall also increases, empirically verifying our Theorem 1 that in expectation, the clean nodes will have stable node representations and thus stable predictions when using the graph convolution operation designed in Eq. 3. As $\beta$ increases from 0.1 to 0.7, the recall also increases without a decrease in precision, demonstrating the stability of our method. When $\beta = 0.9$, only about 10% of edges remain in each iteration. Though the recall decreases slightly, we still achieve consistently high precision. This further indicates the robustness of our model against various chosen hyperparameters.

## N   ADDITIONAL RESULTS OF ABLATION STUDIES

In this section, we provide additional results on ablation studies on OGB-arxiv dataset to investigate the impact of random edge dropping on poison node detection and evaluate the efficacy of our robust training strategy as described in Eq. 6. Following Sec. 5.4, to evaluate the effectiveness of random edge dropping, we replace it with an intuitive method of dropping edges individually, as described in Section 3.2, and obtain a variant named as RIGBD\E. To demonstrate the effectiveness of our robust training strategy, we implement a variant of our model, named RIGBD\R, which simply removes identified candidate poisoned nodes from the dataset. The attack method used is DPGBA. We also implement a variant called RIGBD\RE, which involves individually dropping each edge and eliminate candidate poisoned nodes. The number of iterations for random edge dropping is set to $K = 20$, with a drop ratio of $\beta = 0.5$. We report the ASR and ACC in Fig. 10. The results on on scenarios where a backdoor trigger (subgraph) is linked to a poisoned node by two edges are also provided as a reference. From the figure, we observe: **(i)** In Fig. 10 (a), RIGBD\E and RIGBD\RE achieve comparable and slightly better defense performance than RIGBD\and RIGBD\R. However, in Fig. 10 (b), RIGBD\E and RIGBD\RE fail to degrade the ASR. This demonstrate that although drop edge individually may demonstrate comparable performance in detecting poison nodes when there is only one edge linking a trigger to a target node, it fails to defend effectively against attacks involving two edges linking a backdoor trigger to a target node. In such cases, the intuitive method of dropping each edge individually proves inadequate. **(ii)** In both settings, RIGBD\achieve better defense performance than RIGBD\R. This demonstrates the effectiveness of our robust training strategy in handling cases where some poisoned nodes are not identified.

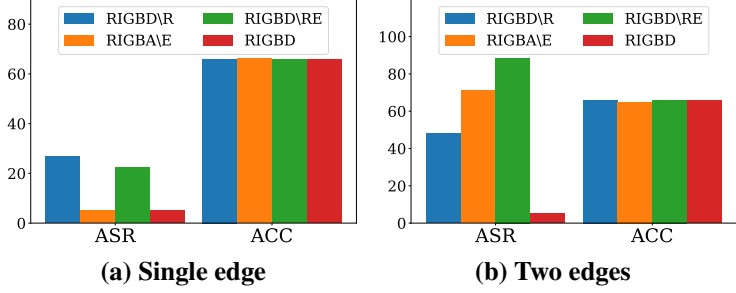

(a) Single edge          (b) Two edges

Figure 10: Ablation study on OGB-arxiv

## O   ADDITIONAL RESULTS OF DEFENSE PERFORMANCE AND ABILITY TO DETECT POISON NODES WITH DIFFERENT NUMBERS OF TRIGGERS

In this section, we conduct experiments to demonstrate how varying numbers of backdoor triggers impact the performance of RIGBD in terms of backdoor defense and poisoned node detection. Specifically, we set the number of triggers to $\{10, 20, 40, 80, 160\}$ for Cora and $\{20, 80, 160, 240, 320\}$ for PubMed. The attack method used is DPGBA. The drop ratio $\beta = 0.5$, and the number of iterations for random edge dropping are set as $K = 10$ for Cora and $K = 20$ for PubMed. The results are

Table 11: Results for defense and poisoned node detection with different numbers of triggers.

| Datasets | Triggers | ASR | Clean ACC | ASR | ACC | Recall | Precision |
|---|---|---|---|---|---|---|---|
| Cora | 10 | 87.82 | 84.81 | 0.05 | 85.56 | 90.0 | 90.0 |
| | 20 | 92.62 | 84.44 | 0.00 | 85.19 | 95.0 | 100.0 |
| | 40 | 93.48 | 83.70 | 0.02 | 85.93 | 97.0 | 97.0 |
| | 80 | 96.52 | 82.22 | 0.10 | 84.81 | 94.0 | 98.4 |
| | 160 | 99.13 | 80.00 | 0.10 | 84.07 | 97.1 | 95.8 |
| PubMed | 20 | 90.73 | 84.88 | 0.01 | 85.21 | 85.0 | 85.0 |
| | 80 | 94.32 | 84.68 | 0.01 | 84.56 | 75.5 | 90.2 |
| | 160 | 96.10 | 84.17 | 0.02 | 84.47 | 82.0 | 90.5 |
| | 240 | 97.46 | 84.14 | 0.01 | 84.16 | 66.2 | 95.8 |
| | 320 | 98.07 | 83.77 | 0.02 | 84.04 | 83.0 | 88.2 |

shown in Table 11. From the table, we observe: **(i)** Though there are fluctuations in the recall and precision in detecting poisoned nodes, the precision always exceeds $85\%$. This indicates that our method of random edge dropping consistently leads to higher prediction variance, helping us precisely identify poisoned nodes regardless of the number of triggers. **(ii)** With the different number of triggers, our method demonstrate consistently superior performance in defense, with ASR close to $0\%$ and ACC comparable to Clean ACC. Notably, when the number of triggers is $240$ for PubMed, the detection recall is around $66\%$, but we still achieve good results in terms of ASR and ACC. This further indicates the stability of our method in training a robust node classifier using Eq. 6, even if some poisoned nodes are not identified. Additional results on the visualization of prediction variance for poisoned nodes and clean nodes with different numbers of triggers are provided in Fig. 11 and Fig. 12.

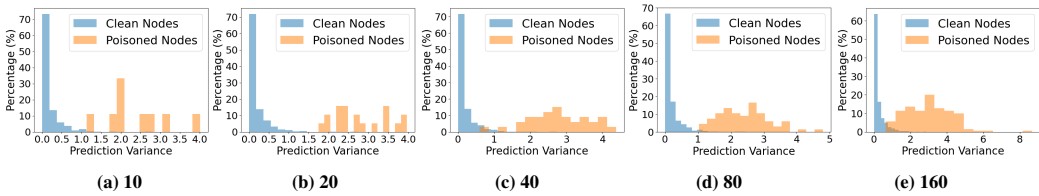

Figure 11: Visualization of prediction variance on Cora dataset with different number of triggers

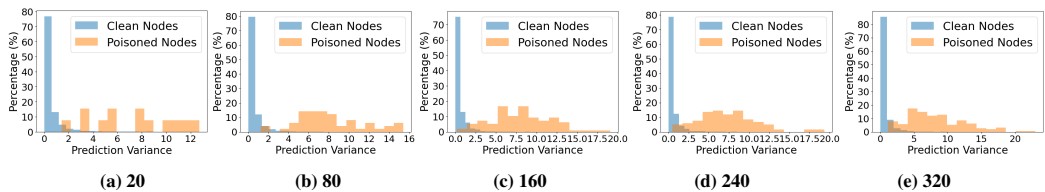

Figure 12: Visualization of prediction variance on PubMed dataset with different number of triggers

# P  ADDITIONAL RESULTS OF DEFENSE PERFORMANCE AND ABILITY TO DETECT POISON NODES WITH TRIGGER SIZE AS 1

In this section, we conduct experiments to investigate how different trigger sizes impact the performance of RIGBD in terms of backdoor defense and poisoned node detection. Specifically, we set the trigger size to 1 and conducted experiments on the Cora, Citeseer, Pubmed, and ogbn-arxiv datasets, keeping all other settings the same as in Section 5.2. The results are shown in Table 12, demonstrating that RIGBD achieves strong performance even with a trigger size of 1.

Table 12: Result of backdoor defense with trigger size as 1

| Dataset | Method | ASR | ACC | Recall | Precision |
|---------|--------|------|-------|--------|-----------|
| Cora | UGBA | 92.53 | 82.22 | - | - |
| | RIGBD | 0.44 | 83.70 | 83.9 | 92.9 |
| Citeseer | UGBA | 95.24 | 74.69 | - | - |
| | RIGBD | 0.00 | 73.20 | 98.7 | 98.7 |
| Pubmed | UGBA | 92.58 | 84.58 | - | - |
| | RIGBD | 0.32 | 84.83 | 94.0 | 97.5 |
| OGB-arxiv | UGBA | 85.29 | 65.38 | - | - |
| | RIGBD | 0.00 | 65.92 | 92.7 | 99.8 |

## Q    REPRODUCIBILITY

Experimental and implementation details are in Sec. 5.1 and Appendix H. We provide a detailed algorithm description in Algo. 1. The code for RIGBD is publicly available at: github.com/zzwjames/RIGBD.

## R    LIMITATIONS AND FUTURE WORK

Our proposed framework aims to defend against graph backdoor attacks during the training phase. We believe it is also important to develop new techniques to defend against graph backdoor attacks during the inference stage, which we leave for future work. As the first work targeting all existing representative graph backdoor attacks, we hope to inspire future development of graph backdoor defense algorithms. Additionally, this paper focuses exclusively on graph-structured data, and it would be interesting to explore extensions to other domains. Given the nature of this work, there are no easily predictable negative social impacts.

