# OpenReview forum: "Robustness Inspired Graph Backdoor Defense"
_ICLR.cc/2025/Conference — ICLR 2025 Oral_

### Official Review · Reviewer_Xv7r · 2024-11-03

**Soundness:** 3
**Presentation:** 3
**Contribution:** 3
**Rating:** 8
**Confidence:** 3

**Summary:**

The paper introduces RIGBD, a novel framework for defending Graph Neural Networks (GNNs) against backdoor attacks. It leverages **random edge dropping** to identify poisoned nodes by detecting high prediction variance, and incorporates a **robust training strategy** to reduce the prediction confidence on these nodes, limiting the impact of triggers. The method is theoretically sound and empirically validated on real-world datasets, showing it can effectively decrease attack success rates while maintaining clean accuracy. Key strengths include scalability and robustness, though the method relies on hyperparameter tuning and may face challenges with indistinguishable poisoned nodes.

**Strengths:**

1. Effective Identification of Poisoned Nodes: The use of random edge dropping as a technique for distinguishing poisoned nodes from clean nodes is innovative and theoretically backed. The method efficiently handles large-scale graphs by reducing the computational burden that would otherwise be associated with processing each edge individually.

2. Stability for Clean Nodes: The approach ensures that clean nodes maintain prediction stability, which prevents sensitivity toward non-poisoned nodes.

**Weaknesses:**

1. Dependency on Hyperparameters: The effectiveness of the random edge drop rate (𝛽) and the number of iterations (𝐾) can vary significantly across different datasets, requiring careful tuning and potentially reducing generalizability. Additionally, there is no theoretical guarantee for selecting the optimal values for these hyperparameters.

**Questions:**

1. In the related work, the SBA method was mentioned. While I understand that SBA may have a low ASR, I am unsure about how your method defends against SBA specifically, as there are no experimental results regarding this method. Could the authors provide a simple experiment to illustrate the defense mechanism against SBA?

2. The paper does not mention any adaptive attacks. Are there any potential ways to design adaptive attacks targeting this method?

---

> ### Author Response · Authors · 2024-11-20
>
> We thank the reviewer for recognizing the valid theoretical and technical details of our work. Below is our point-by-point response to the reviewer's concerns and comments:
>
> **W1: hyperparameter selection**
> * According to our hyperparameter analysis in Section 5.4 and Appendix L, our RIGBD consistently achieves strong performance in defending against backdoor attacks and maintaining clean accuracy across different hyperparameter selections. Specifically, while smaller values of $\beta$ and $K$ may result in poorer recall performance for poisoned node detection compared to larger values, our method still achieves strong performance in reducing ASR, which can be attributed to our robust training strategy.
>
> * Based on our theoretical analysis in Theorem 2 and empirical results in Section 5.4, a larger $\beta$ typically results in better recall performance for poisoned node detection while maintaining stable precision, i.e., minimal impact on clean nodes. This, in turn, leads to improved defense against backdoor attacks and consistent clean accuracy.
>
> * Overall, our RIGBD demonstrates robustness to hyperparameter selection, supported by both theoretical and empirical evidence.
>
> **Q1: experiments on SBA**
>
> Thank you for your suggestion. Following your advice, we conducted experiments on Cora, Pubmed, and OGB-Arxiv using SBA as the attack method. Other settings follow those described in Section 5.1. The results are shown in the table below. It is evident that RIGBD consistently achieves strong performance in defending against backdoor attacks while maintaining clean accuracy across all datasets.
>
> | Dataset    | Method | ASR   | ACC   |
> |------------|--------|-------|-------|
> | Cora       | GCN    | 37.8  | 84.1  |
> |            | RIGBD  | 1.2   | 83.8  |
> | Pubmed     | GCN    | 33.1  | 85.3  |
> |            | RIGBD  | 4.1   | 85.2  |
> | OGB-arxiv  | GCN    | 19.9  | 63.6  |
> |            | RIGBD  | 4.2   | 63.4  |
>
>
> **Q2: adaptive attacks**
>
> We would like to kindly clarify that UGBA and DPGBA are inherently adaptive attacks. Specifically, both methods introduce a learnable trigger generator that generates backdoor triggers based on the target node's attributes. Our RIGBD consistently demonstrates strong performance in defending against both UGBA and DPGBA.

---

> > ### Comment · Reviewer_Xv7r · 2024-11-22
> >
> > Thank you for the reply. I think the reply has addressed most of my concerns. I'll increase my rating to 8.

---

> > > ### Author Response · Authors · 2024-11-24
> > >
> > > We sincerely thank you for recognizing our paper and providing valuable comments.
> > >
> > > Best regards,
> > >
> > > The Authors

---

### Official Review · Reviewer_MqFW · 2024-11-03

**Soundness:** 2
**Presentation:** 3
**Contribution:** 2
**Rating:** 6
**Confidence:** 3

**Summary:**

This paper shows that poisoned nodes exhibit high prediction variance with edge dropping. The authors propose random edge dropping, supported by theoretical analysis, to efficiently and precisely identify poisoned nodes. Furthermore, the authors introduce a robust training strategy to efficiently counteract the impact of the triggers, even if some poisoned nodes remain unidentified.

**Strengths:**

1. The structure of the paper is clear and easy to follow.
2. The paper conducts comprehensive experiments to demonstrate the performance of proposed method.

**Weaknesses:**

1. There are concerns about the efficiency of the method. First, the proposed method requires training the GNN encoder twice, which incurs a large overhead cost, especially in larger datasets such as obg-products[1]. In addition, when calculating the prediction variance, it is necessary to infer K times on the graph, which also brings a large overhead.
2.  The proposed method seems to rely on the homogeneity assumption. In heterophilic graphs, nodes tend to connect to nodes that share different labels with them. Would this result in normal nodes also having higher prediction variance after random edge dropping? It is recommended that the author can give the performance of this method on backdoor attacks in heterophilic graph datasets.

[1] Open Graph Benchmark: Datasets for Machine Learning on Graphs, NeurIPS 2020.

**Questions:**

1. Targeted attack generates attacks on specific nodes and aims to fool GNNs on these target nodes. Can the proposed method defend targeted attack strategies on graphs, such as nettack[1].
2. RIGBD minimizes the predicted confidence of the target class of the poisoned node in Eq. 6, and uses the remaining labeled nodes to calculate the cross entropy loss. If the number of attacked nodes is too large, will this cause insufficient supervision information?

[1] Adversarial attacks on neural networks for graph data, KDD 2018.

---

> ### Author Response · Authors · 2024-11-20
>
> We thank the reviewer for recognizing the valid theoretical and technical details, and comprehensive experiments of our work. Below is our point-by-point response to the reviewer's comments:
>
> **W1: efficiency of RIGBD**
>
> * Compared to vanilla GNN training, our method only requires performing inference K times, followed by retraining a GNN model.
> Specifically, the time complexity for training a vanilla backdoored model is $O(LT(NM^² + |E|M))$, where: $L$ is the number of GNN layers, $N$ is the number of nodes, $E$ is the number of edges, $M$ is the hidden dimension, and $T$ is the number of training epochs.
> In comparison, our framework involves $K$ additional inferences on the entire graph and $T$ epochs for retraining, leading to a time complexity of: $O(L(2T + K)(NM^² + |E|M))$.
> After simplifying this expression, it becomes clear that both complexities belong to the same Big O class, which is: $O(L(NM^² + |E|M))$.
> This means that the time complexity of our framework is asymptotically equivalent to that of training a vanilla backdoored model. The constant factors introduced by the additional steps in our framework do not change the overall order of complexity.
>
> * As demonstrated in hyperparameter analysis in Section 5.4, with a appriorite value of drop ratio $\beta$, though a small value of $K$ will not lead to recall of poison node detection as good as the case with a large $K$, our robust training strategy can already help us achieve a strong performance in defending against backdoor attack, for example, in OGB-arxiv dataset, with $\beta=0.5, K=2$, we degrade the ASR from over 90% to 0.67%. This further enhance the flexibility and efficiency of our RIGBD.
>
> * Compared to RobustGCN, which learns attention weights during the training of a GCN, our framework may be more efficient. Additionally, methods like Randomized Smoothing rely on more epochs of adversarial training and multiple rounds of inference, making them less time-efficient. ABL requires warm-up epochs followed by isolation to split the dataset, which also adds time compared to simple training. The table below shows the running time of RIGBD compared to other baselines on Cora dataset. The results show that our RIGBD is more efficient than most robust learning strategies.
>
> | GCN    | GNNGuard  | RobustGCN | Prune  |OD  |RS  |ABL  |RIGBD  |
> |------------|--------------|-------------|--------------|------------|--------------|-------------|--------------|
> | 7.9s    | 5.7s  | 28.7s | 8.0s  |8.2s  |27.0s  |27.8s  |17.1s  |
>
>
> * **Our RIGBD produces a backdoor-robust model, meaning no further refinement is needed once the model is trained**. In contrast, inference-time defense methods require a defense module to be applied during every inference. When implemented in real-world applications, inference-time defense methods are expected to incur a total defense cost that easily surpasses the cost of training a backdoor-robust model.
>
> **W2: performance on heterophilic graph**
>
> Thank you for your suggestion. Following your advice, we conducted experiments to evaluate the performance of RIGBD on heterophilic graph datasets Wisconsin, Cornell and Texas. The attack method used is DPGBA, with 20 poisoned nodes during training, and other settings follow those described in Section 5.1. The results are presented in the table below. It is evident that RIGBD achieves consistently strong performance in defending against backdoor attacks while maintaining clean accuracy. The recall and precision of poisoned node detection further demonstrate that our method remains robust when applied to heterophilic graphs.
>
> | Dataset    | Methods | Clean ACC | ASR   | Recall | Precision |
> |------------|---------|-----------|-------|--------|-----------|
> | Wisconsin  | GCN     | 48        | 100   | -      | -         |
> |            | RIGBD   | 48        | 0     | 90     | 95        |
> | Cornell    | GCN     | 50.0      | 93.3  | -      | -         |
> |            | RIGBD   | 53.5      | 3.3   | 70     | 100       |
> | Texas      | GCN     | 44.4      | 92.9  | -      | -         |
> |            | RIGBD   | 42.9      | 3.7   | 80     | 70        |
>
> Furthermore, in Line 314, we cited that in Appendix F, we compare the prediction variance of low-degree nodes from heterophilic graphs against poisoned nodes. For your convenience, we present the results in the table below. The results reveal that for heterophilic graphs such as Wisconsin, Cornell, and Texas, the results reveal that prediction variance of poisoned nodes remains significantly higher than that of clean nodes, even for those with low degrees.
>
> | Dataset     | Degree-1 | Degree-2 | Degree-3 | Target Nodes |
> |-------------|----------|----------|----------|--------------|
> | Wisconsin   | 0.64| 1.94     | 1.48     | **6.08**         |
> | Cornell     | 0.79| 1.33     | 1.27     | **5.16**         |
> | Texas       | 0.65| 1.48     | 1.28     | **4.10**         |

---

> > ### Comment · Reviewer_MqFW · 2024-11-21
> > **Thanks for the authors' responses**
> >
> > Thanks for the authors' responses which addressed most of my concerns, I will boost my score to 6. I hope our discussions can be included in the revised version.

---

> > > ### Author Response · Authors · 2024-11-24
> > >
> > > We thank the reviewer for acknowledging our efforts and increasing our score. We will include the discussions in the revised version.
> > >
> > > Best regards,
> > >
> > > The Authors

---

> ### Author Response · Authors · 2024-11-20
>
> **Q1: defense against nettack**
>
> We would like to kindly clarify that the problem definition of Nettack [1] is fundamentally different from that of a backdoor attack. In Nettack, the goal is to modify the node features and graph structure to alter the prediction of a specific target node. In contrast, a backdoor attack aims to poison the GNN model to establish a correlation between the backdoor trigger and the target class. Consequently, any node attached with the backdoor trigger tends to be predicted as the target class, while the backdoor attack simultaneously seeks to maintain clean accuracy on unaffected nodes. Therefore, Nettack falls outside the scope of our paper. We thank the reviewer for pointing this out and will include this discussion in a future version of our paper.
>
>
> **Q2: concerns on insufficient supervision information**
>
> One important goal of a backdoor attack is to maintain clean accuracy on clean nodes in the backdoored model. Thus, **attackers typically avoid introducing too many triggers [2][3][4] into the poisoned graph, as doing so would lead to a significant performance drop in clean accuracy**, making the backdoor attack easily detectable and considered a failure. For example, in UGBA, the attacker introduces only 160 backdoor triggers into the OGB-arxiv dataset, which contains a total of 169,343 nodes.
>
> During a backdoor attack, the model is trained using gradient descent on both poisoned nodes and clean nodes, which allows the model to maintain clean accuracy on clean nodes. This clean accuracy is primarily achieved through training on the clean nodes.
> As shown in the hyperparameter analysis in Section 5.4, our precision in detecting poisoned target nodes consistently exceeds 95%, meaning that only a few clean nodes are filtered out. This ensures that the gradient descent process on clean nodes remains largely unchanged, resulting in the same level of clean accuracy.
>
> [1] Adversarial attacks on neural networks for graph data, KDD 2018.
>
> [2] Graph Backdoor. USENIX Security 2021.
>
> [3] Unnoticeable Backdoor Attacks on Graph Neural Networks. WWW 2023.
>
> [4] Rethinking Graph Backdoor Attacks: A Distribution-Preserving Perspective. KDD 2024.

---

### Official Review · Reviewer_TH4V · 2024-11-04

**Soundness:** 2
**Presentation:** 3
**Contribution:** 3
**Rating:** 8
**Confidence:** 4

**Summary:**

This paper presents Robustness-Inspired Graph Backdoor Defense (RIGBD) to defend against subgraph-based graph backdoor attacks. The authors demonstrate that poisoned nodes exhibit significant prediction variance when anomalous edges linked to them are removed, and propose a robust training strategy to mitigate the effects of backdoor triggers. Their method effectively identifies poisoned nodes, reduces attack success rates, and preserves clean accuracy across various datasets.

**Strengths:**

- RIGBD is specifically designed for graph-structured data, effectively leveraging the behavior of malicious edges linking trigger subgraphs to poisoned nodes. The core insight—that the implanted trigger influences prediction by transmitting malicious information through these edges—is particularly valuable in identifying poisoned nodes.

- This paper introduces an intriguing observation: simply removing backdoor triggers does not necessarily immunize the model against backdoor attacks. This insight opens up new directions for designing graph backdoor defense.

- The paper provides a thorough theoretical analysis that explains why random edge dropping can effectively distinguish poisoned nodes from clean ones.

**Weaknesses:**

- Though experiments demonstrate the superiority of RIGBD against subgraph-based backdoor attacks, its resistance against other forms of graph backdoors, e.g., injecting backdoors in the spectral domain [1], remains unevaluated.

- The method proposed for determining target nodes and labels seems to lack robustness. Specifically, the authors suggest ranking nodes in descending order based on prediction variance after random edge dropping, selecting the top-ranked nodes until a node label differs from the previous one. This approach could be susceptible to outliers or special node characteristics, such as nodes with few neighbors and high class-diversity among them, which may result in large prediction variance under random edge dropping. Such nodes may halt the poisoned node selection process or even lead to incorrect target class identification.

[1] Zhao, Xiangyu, Hanzhou Wu, and Xinpeng Zhang. "Effective Backdoor Attack on Graph Neural Networks in Spectral Domain." IEEE Internet of Things Journal (2023).

**Questions:**

- Would RIGBD be effective for non-subgraph graph backdoor attacks?
- How does RIGBD apply to all-to-all backdoor attacks, where different samples have different target labels?
- How does RIGBD adapt to the special case mentioned in weakness 2?

---

> ### Author Response · Authors · 2024-11-20
>
> We thank the reviewer for recognizing the novelty, valid theoretical and technical details in our work. Below is our point-by-point response to the reviewer's comments:
>
> **W1: experiments on injecting backdoors in the spectral domain**
>
> We thank the reviewer for pointing out the related work; however, there is no code available for this method. Due to time constraints, we are unable to implement their approach during the rebuttal.
>
> **W2 & Q3: robustness of poisoned node selection**
>
> In Line 314, we cited that in Appendix F, we compare the prediction variance of low-degree nodes from both homophilic and heterophilic graphs against poisoned nodes. For your convenience, we present the results in the table below. The results reveal that for heterophilic graphs such as Wisconsin, Cornell, and Texas, the prediction variance gap between low-degree and poisoned nodes is smaller compared to homophilic graphs. However, the prediction variance of poisoned nodes remains significantly higher than that of clean nodes, even for those with low degrees.
>
> | Dataset     | Degree-1 | Degree-2 | Degree-3 | Target Nodes |
> |-------------|----------|----------|----------|--------------|
> | Wisconsin   | 0.64| 1.94     | 1.48     | **6.08**         |
> | Cornell     | 0.79| 1.33     | 1.27     | **5.16**         |
> | Texas       | 0.65| 1.48     | 1.28     | **4.10**         |
> | Cora        | 0.51| 0.66     | 0.56     | **12.70**        |
> | Citeseer    | 0.36| 0.70     | 0.69     | **5.41**         |
> | Pubmed      | 0.67| 0.80     | 0.57     | **10.67**        |
> | Physics     | 0.44| 0.52     | 0.36     | **18.80**        |
> | Flickr      | 0.01| 0.02     | 0.01     | **2.56**         |
> | OGB-arxiv   | 0.57| 0.79|0.67| **10.67**        |
>
> To further address your concerns, we conducted additional experiments to evaluate the performance of RIGBD on heterophilic graphs. The attack method used is DPGBA, with 20 poisoned nodes during training, and other settings follow those described in Section 5.1. The results are presented in the table below. It is evident that RIGBD achieves consistently strong performance in defending against backdoor attacks while maintaining clean accuracy. The recall and precision of poisoned node detection further demonstrate that our method remains robust when applied to heterophilic graphs.
>
> | Dataset    | Methods | Clean ACC | ASR   | Recall | Precision |
> |------------|---------|-----------|-------|--------|-----------|
> | Wisconsin  | GCN     | 48        | 100   | -      | -         |
> |            | RIGBD   | 48        | 0     | 90     | 95        |
> | Cornell    | GCN     | 50.0      | 93.3  | -      | -         |
> |            | RIGBD   | 53.5      | 3.3   | 70     | 100       |
> | Texas      | GCN     | 44.4      | 92.9  | -      | -         |
> |            | RIGBD   | 42.9      | 3.7   | 80     | 70        |
>
> **Q1: non-subgraph graph backdoor attacks**
>
> It is not clear whether the reviewer refers to a backdoor attack with a trigger size of 1, i.e., where the backdoor trigger is a single node instead of a subgraph. If this is the case, in line 479, we cited Appendix O, where we present the results of RIGBD when defending against a backdoor attack with a trigger size of 1. The results consistently demonstrate strong performance in defending against the attack while maintaining clean accuracy. We would be happy to address any additional concerns if the reviewer could kindly provide further clarification on the question.
>
> **Q2: how to defend against all-to-all backdoor attacks (different samples have different target labels)**
>
> Though all-to-all backdoor attacks aim to have different target predictions for different samples, a successful backdoor attack means that for a node with and without a backdoor trigger, the prediction results on the target node will drastically change. Thus, the core idea of RIGBD—that prediction variance is a crucial indicator for identifying poisoned target nodes—still works. To adapt to the case where different target nodes have different target labels, instead of ranking nodes in descending order and selecting the top-ranked nodes until a node label differs from the previous one, we can adjust the method by ranking nodes in descending order and manually setting a threshold to filter out those nodes with prediction variance higher than that threshold as poisoned target nodes. Finally, we can follow the same robust training strategy as in RIGBD. **Overall, when adapting to all-to-all backdoor attacks, we only need to slightly adjust RIGBD by filtering out poisoned target nodes using a manually set threshold**.

---

> > ### Comment · Reviewer_TH4V · 2024-12-02
> > **Reply to the response**
> >
> > Thanks for the response. I have read the comments from other reviewers and will raise my score to support this paper.

---

> > > ### Author Response · Authors · 2024-12-03
> > >
> > > Thank you very much for your kind support. We sincerely appreciate the time and effort you dedicated to reviewing our paper and providing valuable comments.
> > >
> > > Best regards,
> > >
> > > The Authors

---

### Official Review · Reviewer_PrW3 · 2024-11-07

**Soundness:** 3
**Presentation:** 3
**Contribution:** 3
**Rating:** 8
**Confidence:** 2

**Summary:**

This paper proposes a novel defense method against graph backdoor attacks, which is composed of poisoned nodes detection and robust training. In poisoned node detection, this paper first observes that edge dropping significantly influences the prediction of the poisoned nodes and then this has been theoretical verified. After this, this paper proposes to utilize random edge dropping to detect poisoned nodes efficiently based on poisoned GNN model. After detection, the poisoned model will be finetuned by the detected nodes to remove the influence of the backdoor attack.

**Strengths:**

1. The idea is novel and interesting, and is well evaluated empirically and theoretically.
2. In addition to the detection of poisoned node, this paper also proposes robust training to enhance defense performance further. This can safeguard GNN models against different kinds of attacks.
3. A well-written paper, and easy-to-follow.

**Weaknesses:**

1. To verify the effectiveness and generalization of the proposed defense, I suggest author deeply discussing the mechanism of dirty-label backdoor attacks against GNNs.
2.  I find out that the discussion among baselines and the proposed defense is missing, e.g., why the proposed work only performs better on defending DPGBA. This also weakens the contribution of this work. Finally, I find that the baselines are not targeted at graph backdoor attacks, e.g., GNNGuard targets defending adversarial attacks while ABL targets traditional DNNs. It is important to compare with other defense strategies against graph backdoor attacks.
3. I find that the idea of this paper is very similar to the motivation of the influence function which aims to measure the impact of deleting edges and nodes by influence function (in this paper authors use predicted probability) and does not require retraining or fine-tune the poisoned model. I suggest authors discuss and compare with backdoor defense strategies using influence function.

**Questions:**

Please refer to weaknesses.

---

> ### Author Response · Authors · 2024-11-20
>
> We thank the reviewer for recognizing the novelty, sound theoretical and technical details, as well as the extensive experiments in our work. Below is our point-by-point response to the reviewer's comments:
>
> **W1: deep discussion on the mechanism of dirty-label backdoor attack**
>
> As defined in the problem statement in Section 3.1 and described in the introduction, dirty-label graph backdoor attacks involve linking a backdoor trigger to a small set of target nodes, which are then assigned a target class to form a poisoned graph. When a GNN model is trained on a backdoored dataset, it learns to associate the presence of the trigger with the target class. Consequently, during inference, the backdoored model misclassifies test nodes with the trigger attached as belonging to the target class, while still maintaining high accuracy on clean nodes without the trigger. This attack framework is well-studied, with representative works such as GTA [1], UGBA [2], and DPGBA [3]. Empirical results show that with only a small set of poisoned nodes, these attack frameworks can typically achieve over a 90% attack success rate across various datasets. Such attacks pose a significant threat to the application of graph neural networks in real-world scenarios, such as social media networks, where attackers can easily create fake accounts as backdoor triggers and connect them to target user nodes, misleading the model into predicting the user as the target class.
>
>
> **W2: more discussion on DPGBA and baselines**
>
> We have included the discussion among baselines and RIGBD in Appendix H.4. Apologies for not clearly mentioning this in the main pages.
>
>
> *   As mentioned in [3], although GTA [1] and UGBA [2] achieve good attack results, both suffer from the issue that their generated backdoor triggers are outliers compared to clean nodes. Thus, a simple outlier detection method could defend against such attacks. DPGBA [3] proposes generating backdoor triggers that share similar node features with clean nodes, making it difficult to defend against with trivial methods. We believe that future advanced attack methods could enhance attack performance while simultaneously generating triggers capable of bypassing simple defense methods. Therefore, it is of vital importance to propose defense methods like RIGBD to counter such attacks.
> *   Since graph backdoor attack methods [1][2][3] inherently belong to the category of poisoning attacks, we include GNNGuard and RobustGCN as baselines because both have the ability to mitigate the influence of malicious neighbors. This may reduce the impact of backdoor triggers, as these triggers are neighbors of the target nodes. However, their defense performance is not satisfactory, especially when the generated triggers are similar to the target nodes.
> *  The core idea of ABL is that the model quickly learns backdoored data, after which a local gradient ascent technique is used to trap the loss value of each example. Although it was initially designed for DNNs, this defense framework can be easily adapted to GNNs. However, their framework requires manual tuning of the isolation ratio, and a high isolation ratio inevitably leads to a performance drop on clean data when their losses are trapped. This significantly reduces the method's utility in real-world applications, as defenders typically do not know the number of poisoned target nodes.
> * To the best of our knowledge, at the time of submitting our work, there is no published work that specifically aims to defend against dirty-label node-level graph backdoor attacks that involve generating a backdoor trigger and linking it to the target node.
>
>
> [1] Graph Backdoor. USENIX Security 2021.
>
> [2] Unnoticeable Backdoor Attacks on Graph Neural Networks. WWW 2023.
>
> [3] Rethinking Graph Backdoor Attacks: A Distribution-Preserving Perspective. KDD 2024.

---

> ### Author Response · Authors · 2024-11-20
>
> **W3: comparison to influence function**
>
> Regarding the influence function, we appreciate the reviewer for pointing this out. Existing work [4] explores approximating the influence of node removal and edge removal on the change in representation of other nodes. Another work [5] approximates the influence of node/edge removal on the change in model parameters. Since changes in model parameters do not directly reflect changes in node representations and logits prediction, we focus on discussing related work [4] here. However,
>
> (i) their framework and theoretical analysis are based on Simple Graph Convolution rather than GCNs.
>
> (ii) For edge removal, the time complexity of measuring the influence of each edge in the entire graph is $O(n|E||\theta|^2 + |\theta|^3)$, where $|\theta|$ represents the number of model parameters, $|E|$ is the number of edges, and $n$ is the average number of affected nodes after dropping a specific edge. This can be time-consuming for dense graphs and deep graph neural networks. In contrast, the time complexity of our method is $O(LK(NM^2 + |E|M))$, where $L$ is the number of GNN layers, $N$ is the number of nodes, and $M$ is the hidden dimension. Typically, $LKM \ll n|\theta|^2$ and $LNKM^2\ll |θ|^3$. Thus, our method is typically more time-efficient compared to the influence function method.
>
> (iii) As the reviewer mentioned, their method does not require fine-tuning the poisoned model. However, if we adopt the influence function without fine-tuning the poisoned model, each time we perform inference on a new poisoned graph, we would need to check the influence of each edge again, which will significantly degrade the efficiency and utility.
>
> [4] Characterizing the Influence of Graph Elements. ICLR 2023
>
> [5] GIF: A General Graph Unlearning Strategy via Influence Function. WWW 2023

---

> > ### Comment · Reviewer_PrW3 · 2024-11-26
> >
> > Thanks for your response. I will keep my score, and recommend accept.

---

> > > ### Author Response · Authors · 2024-11-30
> > >
> > > We sincerely thank the reviewer for dedicating time to reviewing our paper and providing insightful comments. We greatly appreciate your approval of our work.
> > >
> > > Best regards,
> > >
> > > The Authors

---

### Comment · Area_Chair_vc5H · 2024-11-24

Dear reviewers,

Thanks for serving as a reviewer. As the discussion period comes to a close and the authors have submitted their rebuttals, I kindly ask you to take a moment to review them and provide any final comments.

If you have already updated your comments, please disregard this message.

Thank you once again for your dedication to the OpenReview process.

Best,

Area Chair

---

### Meta-Review · Area_Chair_vc5H · 2024-12-21

**Metareview:**

This paper presents a defense method against graph backdoor attacks, combining poisoned node detection and robust training. It observes that edge dropping significantly affects the prediction of poisoned nodes with theoretical verification. The method uses random edge dropping to efficiently detect poisoned nodes in the GNN model. After detection, the model is finetuned by removing the influence of the detected poisoned nodes.

Strength:
1. Novel methods with comprehensive empirical results and theoretical analysis.

2. Well-written.

Weakness:
The paper's proposed method is only limited to node classification tasks.

In summary, the paper is a good paper for defending backdoor attacks on node classification. All the reviewers show a positive attitude toward this paper and I would like to recommend it as a spotlight paper.

**Additional Comments On Reviewer Discussion:**

The reviewers and authors discuss the methods' complexity, evaluations, etc. These discussions also improve this paper.

---

### Decision · Program_Chairs · 2025-01-22

Accept (Oral)